# ADAPTIVE GENERALIZATION FOR SEMANTIC SEGMENTATION

## ABSTRACT

Out-of-distribution robustness remains a salient weakness of current state-of-the-art models for semantic segmentation. Until recently, research on generalization followed a restrictive assumption that the model parameters remain fixed after the training process. In this work, we empirically study an adaptive inference strategy for semantic segmentation that adjusts the model to the test sample before producing the final prediction. We achieve this with two complementary techniques. Using *Instance-adaptive Batch Normalization* (IaBN), we modify normalization layers by combining the feature statistics acquired at training time with those of the test sample. We next introduce a *test-time training* (TTT) approach for semantic segmentation, Seg-TTT, which adapts the model parameters to the test sample using a self-supervised loss. Relying on a more rigorous evaluation protocol compared to previous work on generalization in semantic segmentation, our study shows that these techniques consistently and significantly outperform the baseline and attain a new state of the art, substantially improving in accuracy over previous generalization methods.

## 1 INTRODUCTION

The current state of the art for semantic segmentation (Long et al., 2015; Chen et al., 2018b) lacks direly in out-of-distribution robustness, *i. e.* when the training and testing distributions are different. Numerous studies have investigated this issue, with a primary focus on image classification (Arjovsky et al., 2019; Bickel et al., 2009; Li et al., 2017a; Torralba and Efros, 2011; Volpi et al., 2018). However, the conclusion from a recent study of *domain generalization* methods (Gulrajani and Lopez-Paz, 2020), spanning more than three years of research, is disillusioning: Empirical Risk Minimization (ERM), which is based on the *i. i. d.* assumption of the training and testing distributions, is still highly competitive. This is in stark contrast to the evident advances in the area of *domain adaptation*, both for image classification (Ben-David et al., 2010; Ganin et al., 2016; Long et al., 2016; Xie et al., 2018) and semantic segmentation (Araslanov and Roth, 2021; Yang and Soatto, 2020; Vu et al., 2019). This setup, however, assumes access to an unlabelled test distribution at training time, whereas in the generalization setting considered here, only one test sample is accessible at inference time, and no knowledge between the subsequent test samples must be shared.

In this work, we study the generalization problem of semantic segmentation from synthetic data (Richter et al., 2016; Ros et al., 2016) through the lens of adaptation. Instead of modifying the model architecture (Pan et al., 2018) or the training process (Chen et al., 2020; 2021; Yue et al., 2019), we enhance the inference procedure with two orthogonal techniques inspired by domain adaptation methods (Araslanov and Roth, 2021; Li et al., 2017b). The first mechanism leverages normalization layers (Ioffe and Szegedy, 2015) in modern convolutional neural networks (CNNs) (He et al., 2016) by integrating feature statistics of the test sample. Expanding upon previously attained conclusions for the image classification task (Schneider et al., 2020), we find that this strategy not only improves the segmentation accuracy, but also the calibration quality of the prediction confidence. Our second contribution is a self-supervised loss, allowing the model to adapt to a single test sample with a few parameter updates. Thirdly, we set up an extensive empirical study following a rigorous evaluation protocol, allowing us to establish that the two techniques are *complementary*. Our study of this one-sample adaptation process reveals a consistent improvement in out-of-distribution robustness over the baseline in *all benchmark scenarios*, and yields a new state of the art in segmentation accuracy, substantially surpassing that of previous work in virtually all considered settings.

## 2 RELATED WORK

Our work contributes to recent research on generalization of semantic segmentation models, and relates to the studies on feature normalization (Pan et al., 2018; Schneider et al., 2020) and test-time training (Sun et al., 2020). While the focus in previous investigations was the training strategy (Yue et al., 2019) and model design (Pan et al., 2018), we exclusively study the test-time inference process here. Yue et al. (2019) augmented the synthetic training data by transferring style from real images. Assuming access to a classification model trained on real images, Chen et al. (2020) regularize the training on synthetic data by ensuring feature proximity of the two models via distillation, and seek layer-specific learning rates for improved generalization. Advancing the distillation technique, Chen et al. (2021) devise a contrastive loss that facilitates model invariance to standard image augmentations. Pan et al. (2018) heuristically add instance normalization (IN) layers to the network. We remark on two limitations of these works that we address here: First, these methods assume access to a *distribution* of real images during training (Chen et al., 2020; 2021; Yue et al., 2019) (as opposed to only for pre-training of the backbone), or require a modification of the network architecture (Pan et al., 2018). Our work requires neither, hence the presented techniques apply even *post-hoc* to the already (pre-)trained models to improve their generalization. Second, as we discuss and address in Sec. 5, the evaluation protocol used by previous studies exhibits a number of shortcomings.

**Normalization.** Batch Normalization (BN; Ioffe and Szegedy, 2015) and other normalization techniques have been increasingly linked to model robustness (Deecke et al., 2019; Huang et al., 2019; Schneider et al., 2020; Wang et al., 2018; Wu and He, 2018). The most commonly used BN, Layer Normalization (LN; Ba et al., 2016), and Instance Normalization (IN; Ulyanov et al., 2016) also affect the model's expressive power, which can be further enhanced by their arrangements (Nam and Kim, 2018; Luo et al., 2019). In a domain adaptation setting, Li et al. (2017b) use source-domain statistics during training while replacing them with target-domain statistics during inference. More recently, Schneider et al. (2020) combine the source and target statistics during inference, but the statistics are weighted depending on the number of samples that these statistics aggregate. Nado et al. (2020) propose using batch statistics during inference from the target domain instead of the training statistics acquired from the source domain. Our comprehensive empirical study complements these results by demonstrating improved generalization of semantic segmentation models.

**Test-time training.** Test-time training (TTT) refers to updating the model parameters also at inference time with a self-supervised loss incurred on a single unlabelled test sample. This technique has been recently applied with success for improving the robustness of image classification models (Sun et al., 2020; Wang et al., 2021a). The design of the self-supervised task is crucial, and the techniques developed for image classification are unsuitable for dense prediction tasks, such as semantic segmentation. Nevertheless, recent work explored such losses in domain adaptation scenarios (Araslanov and Roth, 2021), and a number of other works exploit domain-specific knowledge from medical imaging (Varsavsky et al., 2020) or first-person vision (Cai et al., 2020).

**Setup comparison.** Most of these technically related works (Schneider et al., 2020; Sun et al., 2020; Wang et al., 2021a) focus on the problem of domain adaptation in the context of image classification. They typically assume access to a number of samples (or even all test images) from the target distribution at training time. Our work instead addresses semantic segmentation in the domain generalization setting, which is fundamentally different as it only necessitates a single datum from the test set. In this scenario, simple objectives, such as entropy minimization employed by Tent (Wang et al., 2021a), improve the baseline accuracy only moderately. By contrast, our one-sample adaptation with pseudo labels accounts for the inherent uncertainty in the predictions, which proves substantially more effective, as the comparison to Tent in Sec. 5.3 reveals. Our task is also different from few-shot learning (*e. g.*, (Finn et al., 2017)), where the model may adapt at test time using a small *annotated* set of image samples. No such annotation is available in our setup; our model adjusts to the test sample in a completely unsupervised fashion, requires neither proxy tasks to update the parameters (Sun et al., 2020) nor any knowledge of the test domain.

## 3 INSTANCE-ADAPTIVE BATCH NORMALIZATION

**Batch Normalization** (BN; Ioffe and Szegedy, 2015) has become an inextricable component of modern CNNs (He et al., 2016). Although BN was originally designed for improving training

convergence, there is now substantial evidence that it plays an important role in model robustness (Nado et al., 2020), including domain generalization (Pan et al., 2018). Let $z \in \mathbb{R}^{B,C,H,W}$ denote a feature tensor, with $C$ channels and batch size $B$, produced by a convolutional layer at resolution $H \times W$. We omit the layer indexing, as the following presentation applies to all layers in which BN is applied. At training time, BN first computes the mean and the standard deviation for each of the $C$ channels batch-wise, *i. e.*

$$\mu_c = \tfrac{1}{BHW} \sum_{i,j,k} z_{i,c,j,k} \,, \qquad \sigma_c^2 = \tfrac{1}{BHW} \sum_{i,j,k} (z_{i,c,j,k} - \mu_c)^2. \tag{1}$$

The normalized features $\hat{z}$ follow from applying these statistics,

$$\hat{z}_{i,c,j,k} = (z_{i,c,j,k} - \mu_c)/\sqrt{\sigma_c^2 + \epsilon}. \tag{2}$$

Notably, this process differs from the normalization used at inference time. At training time, every BN layer maintains a running estimate of $\mu_c$ and $\sigma_c$ across the training batches, which we denote here as $\hat{\mu}_c$ and $\hat{\sigma}_c$. At test time, it is an established practice to normalize *w. r. t.* $\hat{\mu}_c$ and $\hat{\sigma}_c$ in Eq. (2), instead of the test-batch statistics, a scheme we refer to as *train BN* (*t*-BN).

**BN and generalization.**   In the context of out-of-distribution generalization, the running statistics $\hat{\mu}_c$ and $\hat{\sigma}_c$ derive from the source data and can be substantially different had they been computed using the target images. This discrepancy is generally known as the *covariate shift* problem. Domain adaptation methods, which assume access to the (unlabelled) target distribution, often alleviate this issue with a technique typically referred to as Adaptive Batch Normalization (AdaBN; Li et al., 2017b). The key idea behind the method is to simply replace the source running statistics with those of the target. Instead of alternating BN layers between the training and testing modes, recent work (Nado et al., 2020) studies *prediction-time BN* (*p*-BN), which maintains BN layers in "training mode" also at inference time. That is, *p*-BN replaces the running statistics from the training time, $\hat{\mu}_c$ and $\hat{\sigma}_c$, with the statistics $\mu_c$ and $\sigma_c$ of the current test batch. Such a seemingly innocuous change is shown to benefit model robustness for image classification (Nado et al., 2020; Schneider et al., 2020).

In contrast to AdaBN and *p*-BN, which utilize either the whole target distribution or a number of samples, our study of model generalization assumes only a single target example to be available — the one that our model receives as the input at inference time. A viable alternative to AdaBN and *p*-BN is to compute the statistics per sample, which amounts to replacing BN layers with Instance Normalization (IN) layers (Ulyanov et al., 2016) after model training. However, this may cause another extreme scenario for covariate shift, since such replacement may significantly interfere with the statistics of the activations in the intermediate layers with which the network was trained. Moreover, previous work showed that IN layers hurt the discriminative power of the model and can improve model robustness only in combination with BN (Pan et al., 2018), thereby requiring changes in the model architecture.

**Instance-adaptive batch normalization.**   Following Schneider et al. (2020), we compute a weighted average of the running statistics from training on source data and the target statistics, which in contrast to Schneider et al. (2020) always derives from a single sample in our work. Let $\hat{\mu}_c^{(s)}$ and $\hat{\sigma}_c^{(s)}$ denote the running estimate of the mean and the standard deviation at training time on the source data (for feature channel $c$). Similarly, define $\mu_c^{(t)}$ and $\sigma_c^{(t)}$ as the mean and the standard deviation computed from a feature tensor $z^{(t)} \in \mathbb{R}^{C,H,W}$, coming from a *single* target sample:

$$\mu_c^{(t)} = \tfrac{1}{HW} \sum_{j,k} z_{c,j,k}^{(t)} \,, \qquad \sigma_c^{(t)^2} = \tfrac{1}{HW} \sum_{j,k} (z_{c,j,k}^{(t)} - \mu_c^{(t)})^2. \tag{3}$$

At inference time, we compute the new mean and standard deviation, $\mu_c^{(*)}$ and $\sigma_c^{(*)}$, as follows:

$$\mu_c^{(*)} = \alpha \mu_c^{(t)} + (1-\alpha)\hat{\mu}_c^{(s)} \,, \qquad \sigma_c^{(*)^2} = \alpha \sigma_c^{(t)^2} + (1-\alpha)\hat{\sigma}_c^{(s)}. \tag{4}$$

We then use $\mu_c^{(*)}$ and $\sigma_c^{(*)}$ in place of $\mu_c$ and $\sigma_c$ in Eq. (2) to normalize the features. Note that this does not affect the behaviour of the BN layers at training time and applies only at test time. Since this approach combines the inductive bias coming in the form of the running statistics from the source domain with statistics extracted from a single test instance, we refer to this technique as *Instance-adaptive Batch Normalization* (IaBN).

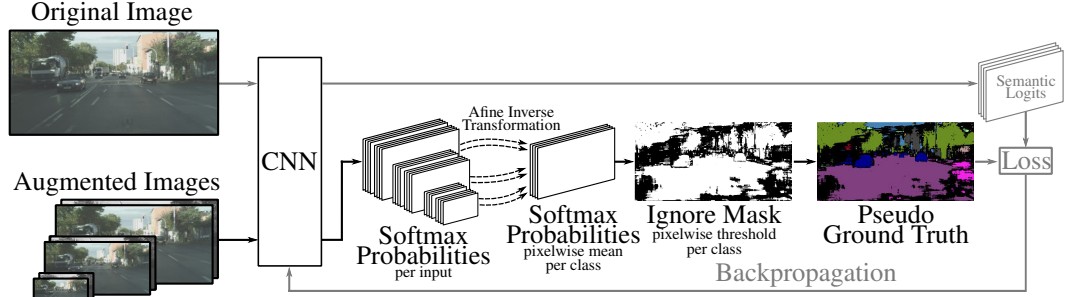

Figure 1: *Overview of the one-sample adaptation process.* Based on a single test sample, we create a batch of images by augmenting the original input with multi-scale versions. Furthermore, we add a horizontally flipped and grayscaled version for every scale. In order to project the resulting output from every version back to the original image, we apply the respective inverse affine transformation to every prediction. Afterwards, we average the predicted softmax probabilities and create a pseudo label using a class-dependent confidence threshold. We update the model parameters by minimizing the cross-entropy loss *w. r. t.* the pseudo label and repeat this process for a small number of iterations, $N_t$, before producing the final prediction. The updated model is then discarded.

**Discussion.** Setting $\alpha = 0$ in Eq. (4) defaults to the established procedure, *t*-BN, and uses only the running statistics from training on the source domain at inference time. Conversely, $\alpha = 1$ corresponds to Instance Normalization (Ulyanov et al., 2016) or, equivalently, to the *p*-BN strategy (Nado et al., 2020) with a batch size of $1$. This is in contrast to (Nado et al., 2020), where $\mu_c^{(t)}$ and $\sigma_c^{(t)}$ are averaged over a batch of target images, whereas we rely on a single test sample in this work. Our experiments in Sec. 5.1 expand upon previous analyses (Schneider et al., 2020) with an extensive emperical study of this normalization strategy for semantic segmentation.

Note that Batch Instance Normalization (Nam and Kim, 2018) used a similar weighting approach for adaptive stylization. However, $\alpha$ was a *training* parameter, whereas we use $\alpha$ only for model selection using a validation set.

In Sec. 5 we empirically verify that IaBN yields a consistent boost of the segmentation accuracy in the out-of-distribution scenario. Perhaps more surprisingly, we also find significant improvements in model calibration (in terms of the expected calibration error, ECE (Naeini et al., 2015)). We leverage this further in devising a self-supervised approach for test-time training of the segmentation model.

## 4 LEARNING FROM A SINGLE SAMPLE

So far, we assumed that the parameters of the segmentation model remain fixed at inference time. However, both from the perspective of adaptive systems and their biological counterparts, this assumption seems implausible (Thrun, 1998; Widmer and Kubat, 1996). Here, we allow the model to update its parameters. Note that our setup is distinct from the domain adaptation scenario, in contrast to Wang et al. (2021a), since we discard the updated parameters when processing the next sample.

Our approach, visualized in Fig. 1, uses data augmentation as a method to create mini-batches of images for each test sample. Based on the original test image, we first create a set of augmented images by multi-scaling, horizontal flipping, and grayscaling. These augmented images are used to form a mini-batch, which is fed through the CNN. We project the produced softmax probabilities from the model back to the original pixels using the inverse affine transformations, and denote the result as $m_{i,:,:,:}$ for every sample $i$ in the mini-batch. This allows the model to have multiple predictions for one pixel. We then compute the mean $\bar{m}$ of these softmax probabilities along the mini-batch dimension $i$ for class $c$, as

$$\bar{m}_{c,j,k} = \frac{1}{N} \sum_i m_{i,c,j,k}.$$ (5)

Using hyperparameter $\psi \in (0,1)$, we compute a threshold value $t_c$ from the maximum probability of every class to yield a class-dependent threshold $t_c$:

$$t_c = \psi \cdot \max(\bar{m}_{c,:,:}).$$ (6)

We finally extract the class $c_{j,k}^*$ with the highest probability for every pixel by a simple $\arg\max$ operation:

$$c_{j,k}^* = \arg\max(\bar{m}_{:,j,k}). \tag{7}$$

We ignore low-confidence predictions using our class-dependent threshold $t_c$. Specifically, all pixels with a softmax probability below the threshold are set to an ignore label, while the remaining pixels use the dominant class $c_{j,k}^*$ as the pseudo label $u_{j,k}$,

$$u_{j,k} = \begin{cases} c_{j,k}^*, & \text{if } \max(\bar{m}_{:,j,k}) \geq t_{c_{j,k}^*} \\ \text{ignore}, & \text{else.} \end{cases} \tag{8}$$

This generated pseudo ground truth $u$ for the original test image is used to fine-tune the model for $N_t$ iterations using the cross-entropy loss and gradient descent. We determine all hyperparameters, *i.e.* resolution of the scales, threshold $\psi$, number of iterations $N_t$, and learning rate $\eta$, based on a development dataset. After the one-sample adaptation process, we produce a single final prediction using the updated model weights. To process the next test sample, we reset these weights to their initial value, hence the model obtains no knowledge about the complete target data distribution.

## 5 EXPERIMENTS

Previous studies (Chen et al., 2020; 2021; Pan et al., 2018; Yue et al., 2019) on domain generalization for semantic segmentation used a number of evaluation schemes. For example, Chen et al. (2020; 2021); Pan et al. (2018) used only a single target domain, Cityscapes (Cordts et al., 2016), for testing. However, like other research datasets, Cityscapes (Cordts et al., 2016) is carefully curated (*e. g.*, the same camera hardware and country was used for the image capture), hence only partially represents the visual diversity of the world. A more comprehensive approach by Yue et al. (2019) considered a number of target domains and used the average accuracy across those as a generalization metric. However, the model was separately selected for every target domain based on a *different* validation set. As a result, the average accuracy across the tested domains represents the expected accuracy of a model ensemble. In this work, in contrast, we aim to improve the accuracy of a single model.

To that end, we propose a revised evaluation protocol to quantify the generalization ability of a *single* model. We consider a practical scenario in which a supplier prepares a model for a consumer without *a-priori* knowledge on where this model may be deployed. On the supplier's side, we assume access to two data distributions for model training and validation, the *source data* and the *development set*. We assess the generalization ability of the model yielded by the validation process on three qualitatively distinct *target sets*. The average accuracy across these sets provides an estimate of the expected model accuracy for its out-of-distribution deployment on the consumer's side. Next, we concretize the datasets used in this study, limited to traffic scenes for compatibility with previous work (Chen et al., 2021; Pan et al., 2018; Yue et al., 2019), and detail the dataset specifics in Appendix F.

**Source data.**  We train our model on the training split of two synthetic datasets (mutually exclusive) with low-cost ground truth annotation: GTA (Richter et al., 2016) and SYNTHIA (Ros et al., 2016). Importantly, these datasets exhibit visual discrepancy (*i. e.* domain shift) *w. r. t.* the real imagery, from which our model needs to generalize.

**Development set.**  For model selection and hyperparameter tuning, we use the validation set of WildDash (Zendel et al., 2018). In our scenario, the development set is understood to be of limited quantity, owing to its more costly annotation compared to the source data. In contrast to the training set, however, it bears closer visual resemblance to the potential target domains.

**Multi-target evaluation.**  Following model selection, we evaluate the single model on three target domains comprising the validation sets from three datasets of street scenes: Cityscapes (Cordts et al., 2016), BDD (Yu et al., 2020), and IDD (Varma et al., 2019). The choice of these test domains stems from a number of considerations, such as the geographic origin of the scenes (Cityscapes, BDD, and IDD were collected in Germany, North America, and India, respectively). Geographic distinction as well as substantial differences in data acquisition (*e. g.*, camera properties) of these datasets bring together an assortment of challenges for the segmentation model at test time. Since the deployment site of our model is unknown, we assume a uniform prior over the target domains as our test distribution. Under this assumption, a simple average of the mean segmentation accuracy across our target domains yields the expected model accuracy.

Table 1: *(a) Segmentation accuracy using IaBN.* We report the mean IoU (%) on three target domains (Cityscapes, BDD, IDD) across both backbones. *t*-BN denotes train BN (Ioffe and Szegedy, 2015), while *p*-BN refers to prediction-time BN (Nado et al., 2020). *(b) ECE (%) for IaBN and MC-Dropout (Gal and Ghahramani, 2016).* We report scores for three target domains (Cityscapes, BDD, IDD) across both backbones. We trained the networks on GTA in both cases (*cf.* supplemental material for results with SYNTHIA training).

| (a) | | | | | (b) | | | |
|---|---|---|---|---|---|---|---|---|
| Method | IoU (%, ↑) | | | | Method | ECE (%, ↓) | | |
| | CS | BDD | IDD | Mean | | CS | BDD | IDD |
| ResNet-50 | | | | | ResNet-50 (Baseline) | 37.28 | 35.61 | 27.73 |
| w/ *t*-BN | 30.95 | 28.52 | 32.78 | 30.75 | w/ IaBN | 30.57 | 30.94 | 26.90 |
| w/ *p*-BN | **37.71** | 31.67 | 30.85 | 33.41 | w/ MC-Dropout | 30.29 | 29.80 | 24.17 |
| w/ IaBN *(ours)* | 37.54 | **32.79** | **34.21** | **34.85** | w/ MC-Dropout + IaBN | **25.50** | **27.36** | **22.62** |
| ResNet-101 | | | | | ResNet-101 (Baseline) | 35.24 | 33.74 | 27.28 |
| w/ *t*-BN | 32.90 | 32.54 | 30.36 | 31.93 | w/ IaBN | 26.12 | 28.89 | 23.98 |
| w/ *p*-BN | 39.88 | 34.30 | 33.05 | 35.74 | w/ MC-Dropout | 31.30 | 29.95 | 25.15 |
| w/ IaBN *(ours)* | **42.17** | **35.40** | **33.52** | **37.03** | w/ MC-Dropout + IaBN | **24.44** | **28.68** | **23.32** |

To compare to previous works, we also evaluate on Mapillary (Neuhold et al., 2017). Mapillary does not publicly disclose the geographic origins of individual samples, hence is unsuitable to identify a potential location bias acquired by the model from the training data. This is possible in our proposed evaluation protocol, since the geographic locations from Cityscapes, BDD, and IDD do not overlap.

**Implementation details.** We implement our framework[1] in PyTorch (Paszke et al., 2019). Following (Pan et al., 2018), our baseline model is DeepLabv1 (Chen et al., 2015) without CRF post-processing, but the reported results also generalize to more advanced architectures (see Appendix D). We use ResNet-50 and ResNet-101 (He et al., 2016) pre-trained on ImageNet (Deng et al., 2009) as backbone. We minimize the cross-entropy loss with an SGD optimizer and a learning rate of 0.005, decayed polynomially with the power set to 0.9. All models are trained on the source domains for 50 epochs with batch size, momentum, and weight decay set to 4, 0.9, and 0.0001, respectively. For data augmentation, we compute crops of random size (0.08 to 1.0) of the original image size, apply a random aspect ratio (3/4 to 4/3) to the crop, and then resize the result to $512 \times 512$ pixels. Furthermore, we use random horizontal flipping, color jitter, random blur, and grayscaling. We train our models with SyncBN (Paszke et al., 2019) on two NVIDIA GeForce RTX 2080 GPUs.

## 5.1 EVALUATING IaBN

For both source domains (GTA, SYNTHIA) in combination with all main target domains (Cityscapes, BDD, IDD), we investigate the influence of $\alpha$ on the IoU. Setting an optimal $\alpha$ for every target domain is infeasible in domain generalization as the target domain during inference is unknown. Instead, we choose the optimal $\alpha$ in steps of $0.1$ based on the IoU on the development set of WildDash. For the ResNet-50 backbone, we attain the highest validation IoU for both training datasets with $\alpha = 0.1$ (see Fig. 4 in Appendix B). Fixing this optimal $\alpha$, we proceed with evaluating our model on the target domains. Table 1 reports the segmentation accuracy with the optimal $\alpha$. For comparison, we also test our models with the two boundary values of $\alpha = 0$ and $\alpha = 1$, corresponding to *t*-BN and *p*-BN (*cf.* Sec. 3), respectively. In Table 1(a) we report IoU scores for both backbones on generalization from GTA to Cityscapes, BDD, and IDD and compare the accuracy of the target domains with *t*-BN and *p*-BN. Remarkably, IaBN improves the mean IoU not only of the *t*-BN baseline (*e. g.*, by 4.1% IoU with ResNet-50), which represents an established evaluation mode, but also over the more recent *p*-BN (Nado et al., 2020). This improvement is consistent across the board, *i. e.* irrespective of the backbone architecture and the target domain tested. Furthermore, we found that the calibration of our models, in terms of the expected calibration error (ECE; Naeini et al., 2015), also improves. As shown in Table 1(b), not only does IaBN substantially enhance the baseline, but is even competitive with the commonly used MC-Dropout method (Gal and Ghahramani, 2016). Rather surprisingly, IaBN exhibits a complementary effect with MC-Dropout: the calibration of the predictions improves even further when both methods are used jointly.

---

[1]Our code and pre-trained models will be made publicly available under the Apache License.

Table 3: *Mean IoU (%) with TTA (Simonyan and Zisserman, 2015) and our Seg-TTT.* We report scores across both source domains (GTA, SYNTHIA) and three target domains (Cityscapes, BDD, IDD) for both backbones.

| Method | Source: GTA | | | | Source: SYNTHIA | | | |
|---|---|---|---|---|---|---|---|---|
| | CS | BDD | IDD | Mean | CS | BDD | IDD | Mean |
| ResNet-50 (w/ IaBN) | 37.54 | 32.79 | 34.21 | 34.85 | 36.14 | 26.66 | 26.37 | 29.72 |
| w/ TTA | 42.56 | 37.72 | 37.98 | 39.42 | 39.67 | 32.10 | 30.46 | 34.08 |
| w/ Seg-TTT *(ours)* | **45.13** | **39.61** | **40.32** | **41.69** | **41.60** | **33.35** | **31.22** | **35.39** |
| ResNet-101 (w/ IaBN) | 42.17 | 35.40 | 33.52 | 37.03 | 38.01 | 28.66 | 27.28 | 31.32 |
| w/ TTA | 44.37 | 38.49 | 38.35 | 40.40 | 39.91 | 32.68 | 30.04 | 34.21 |
| w/ Seg-TTT *(ours)* | **46.99** | **40.21** | **40.56** | **42.59** | **42.32** | **33.27** | **31.40** | **35.66** |

Table 2: *Mean IoU (%) comparison of IaBN to alternative normalization strategies: SN (Luo et al., 2019) and BIN (Nam and Kim, 2018).*

| Method | CS | BDD | IDD | Mean |
|---|---|---|---|---|
| SN | 31.75 | **33.60** | 31.60 | 32.32 |
| BIN | 34.57 | 32.68 | 30.22 | 32.49 |
| IaBN *(ours)* | **37.54** | 32.79 | **34.21** | **34.85** |

**Comparison to other related work.** We additionally compare IaBN to alternative normalization strategies proposed in the literature: Batch-Instance Normalization (BIN; Nam and Kim, 2018) and Switchable Normalization (SN; Luo et al., 2019). Although both of these techniques may appear technically similar to our IaBN, these approaches were developed for different purposes. SN was only shown to improve in-domain accuracy, while BIN tackles domain adaptation for image classification. Our work studies domain generalization. Furthermore, both methods modify the model architecture before training, while IaBN works with any pretrained semantic segmentation model. We implemented both in our segmentation model based on the ResNet-50 backbone. We trained these approaches on GTA in an identical setup as IaBN. From results in Table 2, we observe that IaBN outperforms both BIN and SN by a significant margin in terms of mean IoU of the target domains.

## 5.2 EVALUATING SEG-TTT

In addition to comparing our Seg-TTT to standard inference, we test our models against Test-Time Augmentation (TTA) (Simonyan and Zisserman, 2015) as a stronger baseline. TTA augments the test samples with their flipped and grayscaled version on multiple scales and averages the predictions as the final result. For Seg-TTT, we use horizontal flipping and grayscaling with factor scales of (0.25, 0.5, 0.75) *w. r. t.* the original image resolution. We study the relative importance of these augmentation types in Appendix C. Based on the validation set WildDash, we set threshold $\psi = 0.7$, use $N_t = 10$ iterations and a learning rate $\eta = 0.05$. We only train the layers conv4_x, conv5_x, and the classification head as we did not observe any benefits from updating all model parameters. Furthermore, this reduces runtime due to not backpropagating through the whole network. We investigate this choice as part of the runtime-accuracy trade-in in Sec. 5.3. In Table 3, we show IoU scores for both source domains (GTA, SYNTHIA) and three target domains (Cityscapes, BDD, IDD) across both backbones. Even though TTA improves the baseline (*e. g.*, by 3.37% IoU with ResNet-101 using GTA), our proposed Seg-TTT still outperforms it by a clear and consistent margin of 2.19% IoU on average. This observation aligns well with our reported ECE scores in Table 1(b) to demonstrate that Seg-TTT further exploits the calibrated confidence of our predictions to yield reliable pseudo labels for adapting the model for a particular test sample.

## 5.3 COMBINING IABN AND SEG-TTT

Both IaBN and Seg-TTT are orthogonal techniques and can be used jointly. Surprisingly, we find the techniques *mutually complementary*. We combine IaBN and Seg-TTT in our model and compare with state-of-the-art domain generalization methods in Table 4. While most of the other methods report their results on weaker baselines, we show consistent improvements even over a substantially stronger baseline. Our single model with IaBN and Seg-TTT even outperforms the model ensemble approach of Yue et al. (2019) on most benchmarks (*e. g.*, by 13.37% and 9.44% on GTA to Mapillary

Table 4: *Mean IoU (%) comparison to state-of-the-art domain generalization methods* for both source domains (GTA, SYNTHIA) as well as three target domains (Cityscapes, Mapillary, BDD). We compare to IBN-Net (Pan et al., 2018), Yue et al. (2019), ASG (Chen et al., 2020) and CSG (Chen et al., 2021). In-domain training to obtain the upper bounds uses our baseline DeepLabv1 and follows the same schedule as with the synthetic datasets. ($^{‡}$) and ($^{†}$) denote the use of FCN (Long et al., 2015) and DeepLabv2 (Chen et al., 2018a) architectures, respectively.

| | Method | Backbone: ResNet-50 | | | Backbone: ResNet-101 | | |
| --- | --- | --- | --- | --- | --- | --- | --- |
| | | CS | Mapillary | BDD | CS | Mapillary | BDD |
| | In-domain Bound | 71.23 | 58.39 | 58.53 | 73.84 | 62.81 | 61.19 |
| *GTA* | No Adapt IBN-Net | 22.17 / 29.64 ↑7.47 | – | – | – | – | – |
| | No Adapt Yue et al. (2019)$^{‡}$ | 32.45 / 37.42 ↑4.97 | 25.66 / 34.12 ↑8.46 | 26.73 / 32.14 ↑5.41 | 33.56 / 42.53 ↑8.97 | 28.33 / 38.05 ↑9.72 | 27.76 / 38.72 ↑10.96 |
| | No Adapt ASG$^{†}$ | 25.88 / 29.65 ↑3.77 | – | – | 29.63 / 32.79 ↑3.16 | – | – |
| | No Adapt CSG$^{†}$ | 25.88 / 35.27 ↑9.39 | – | – | 29.63 / 38.88 ↑9.25 | – | – |
| | No Adapt Ours | 30.95 / **45.13** ↑14.18 | 34.56 / **47.49** ↑12.93 | 28.52 / **39.61** ↑11.09 | 32.90 / **46.99** ↑14.09 | 36.00 / **47.49** ↑11.49 | 32.54 / **40.21** ↑7.67 |
| *SYNTHIA* | No Adapt Yue et al. (2019)$^{‡}$ | 28.36 / 35.65 ↑7.29 | 27.24 / 32.74 ↑5.50 | 25.16 / 31.53 ↑6.37 | 29.67 / 37.58 ↑7.91 | 28.73 / 34.12 ↑5.39 | 25.64 / **34.34** ↑8.70 |
| | No Adapt Ours | 31.83 / **41.60** ↑9.77 | 33.41 / **41.21** ↑7.80 | 24.30 / **33.35** ↑9.05 | 37.25 / **42.32** ↑5.07 | 36.84 / **41.20** ↑4.36 | 29.32 / 33.27 ↑3.95 |

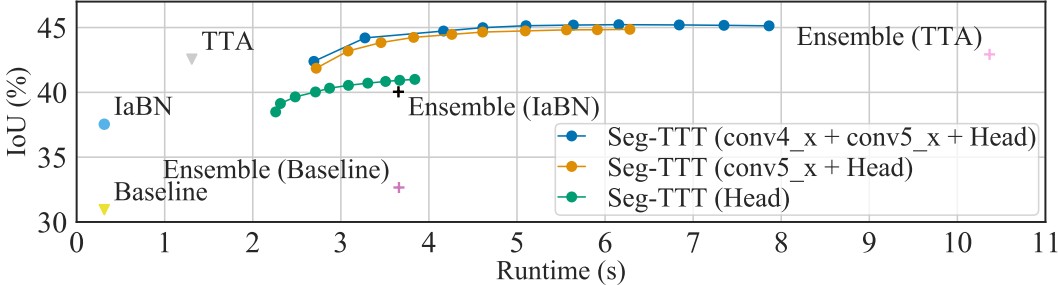

Figure 2: *Runtime-accuracy comparison on GTA $\rightarrow$ Cityscapes generalization.* The curves trace Seg-TTT iterations, *i. e.* the first point corresponds to $N_t = 1$, while the last shows $N_t = 10$. While Seg-TTT increases the inference time of the baseline and TTA for the sake of improved accuracy, it is still more efficient and accurate than model ensembles of 10 networks. The choice of the layers for Seg-TTT updates (the naming follows He et al. (2016)) further provides a favorable runtime-accuracy trade-off. Runtimes are computed on a single NVIDIA GeForce RTX 2080 GPU.

with ResNet-50 and ResNet-101, respectively). Recall that ASG (Chen et al., 2020) and CSG (Chen et al., 2021) (as well as Yue et al. (2019)) require access to a distribution of real images for training, while IBN-Net (Pan et al., 2018) modifies the model architecture. Our approach requires neither, alters only the inference procedure, yet outperforms these methods on most benchmarks substantially. As in the previous discussion, the improvement over the baseline is consistent, regardless of backbone architecture or source data.

**Runtime-accuracy trade-off.** We investigate the influence of the number of iterations required to adapt to a single sample during Seg-TTT. Fig. 2 plots IoU scores for Cityscapes using the ResNet-50 backbone trained on GTA (Appendix C provides further numerical comparison). As a widely adopted baseline, we also train a model ensemble comprising 10 DeepLabv1 networks (as in Seg-TTT), initialized with a random seed (Hansen and Salamon, 1990). Note that TTA, Seg-TTT, and the ensemble use IaBN for a fair comparison, and we also test the ensemble with TTA. We observe that although Seg-TTT increases the accuracy of the baselines at the expense of higher test-time latency,

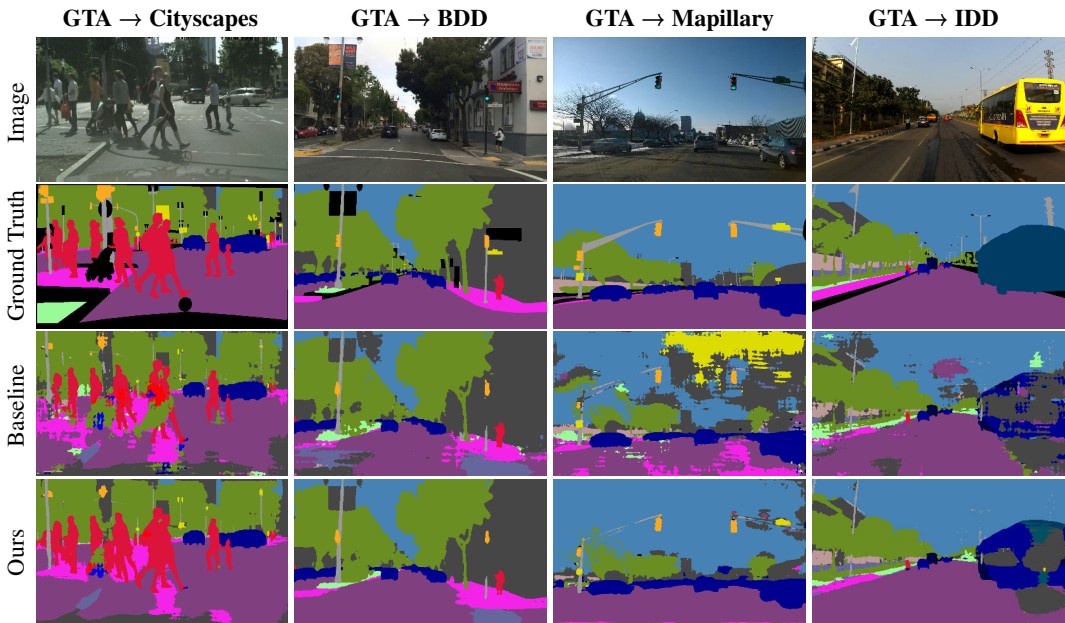

Figure 3: *Qualitative semantic segmentation results for the generalization from GTA to Cityscapes, BDD, Mapillary, and IDD for the ResNet-50 backbone. The input image, ground truth, prediction of the baseline model, and prediction of the proposed combination of IaBN and Seg-TTT are shown.*

it is still more efficient and more accurate than the model ensembles. Furthermore, Seg-TTT offers a practical accuracy-runtime trade-off by means of using fewer update iterations, or updating fewer upper network layers. While the top-accuracy variant of Seg-TTT may not be suitable for real-time applications yet, it may still be valuable in other important domains, such as medical imaging, where high accuracy is desirable even at the cost of increased latency. For real-time needs, IaBN alone boosts the accuracy of the baseline significantly without any computational overhead.

**Comparison to Tent (Wang et al., 2021a).** Like Seg-TTT, Tent also relies on test-time training. However, different from constructing the pseudo labels based on well-calibrated predictions in our Seg-TTT, Tent simply minimizes the entropy of a single-scale prediction. Tent also limits the adaptation to updating only the BN parameters, whereas our Seg-TTT extends this process to convolutional layers. To demonstrate these advantages, we trained HRNet-W18 (Wang et al., 2021b) on GTA and compare the IoU on Cityscapes to the equivalent configuration of Tent. While Tent reaches 36.4% with 10 update iterations by adapting the model to a single image, IaBN alone already outperforms it substantially with a single forward pass (40.0%), and reaches 44.1% with 10 Seg-TTT update iterations.

**Qualitative results.** In Fig. 3 we visualize qualitative segmentation results from our joint model with IaBN and Seg-TTT for generalization from GTA to Cityscapes, BDD, Mapillary, and IDD. We observe a clearly perceivable improvement over the baseline, especially in terms of consistency *w. r. t.* the image boundaries. Appendix G illustrates further results and offers a discussion of failure cases.

## 6 CONCLUSION

We presented a study of an adaptive inference process for improving out-of-distribution robustness of semantic segmentation models. The accuracy improvement demonstrated in the study is surprisingly substantial, despite no changes to the training process or the model architecture, unlike in previous works (Chen et al., 2020; Yue et al., 2019). Considering the simplicity of the studied approach, yet its significant empirical benefits, we hope that such test-time adaptive strategies can inspire follow-up work on improving out-of-distribution generalization of models in other research subfields, potentially extending to other application domains and dense prediction tasks, such as panoptic segmentation, or monocular depth prediction. In future work, we are excited to explore more effective self-supervised loss functions, as well as more efficient test-time adjustments to the model parameters.

**Ethics Statement.** Although increased out-of-distribution robustness can be potentially life-saving in self-driving scenarios, semantic segmentation models can be also deployed with malicious intent, such as unauthorized surveillance. However, semantic segmentation models are still subject to fundamental research with generally low levels of technology readiness (1 to 3) to date and cannot discriminate on the object level (in contrast to, *e. g.*, instance segmentation). These reasons make such unwelcome deployment unlikely without any expert knowledge that we commit not to provide. Moreover, the accuracy of semantic segmentation approaches, while impressive, has not yet reached levels where these methods should be uncritically deployed.

**Reproducibility Statement.** To facilitate reproducibility, we provide our implementation to the reviewers, and commit to releasing it publicly upon acceptance. We have also discussed even trivial, but crucially useful training details of our baseline in-depth in Appendix A and at the start of Sec. 5.

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

## A    BASELINE

We found a number of training details to be crucial for obtaining a highly competitive baseline, *i. e.* a model without our IaBN and Seg-TTT components. Among them is using heavy data augmentation. Recall from Sec. 5 that we used random horizontal flipping, multi-scale cropping with a scale range of $[0.08, 1.0]$, as well as photometric image perturbations:[2] color jitter, random blur, and grayscaling. Color jitter, applied with probability 0.5, perturbs image brightness, contrast, and saturation using a factor sampled uniformly from the range $[0.7, 1.3]$. We use a different range of $[0.9, 1.1]$ for the hue factor. We randomly blur the image using a Gaussian kernel with the standard deviation sampled from $[0.1, 2.0]$. Additionally, we convert the image to grayscale with a probability of $0.1$. Furthermore, we also found that the polynomial decay schedule we used for the learning rate, as well training for at least 50 epochs (for both GTA and SYNTHIA) are essential to achieve a high baseline accuracy. Note that we only used WildDash as the development set to tune these training details. We also experimented with higher input resolution and a larger batch size, but did not observe a significant improvement, yet a drastic increase in the computational overhead.

**On importance of the baseline.**    We note that the reported accuracy from previous work in Table 4 exhibits an inconsistency *w. r. t.* the choice of the model architectures. In particular, Yue et al. (2019) use an FCN, yet outperform other domain generalization approaches with DeepLabv1 (Pan et al., 2018) and DeepLabv2 (Chen et al., 2020; 2021) considerably, as shown in Table 4. This is a regrettable consequence of inconsistent training schedules used in previous works that proved difficult to reproduce. For example, at the time of submission, Yue et al. (2019) have not released their code despite the promise;[3] Pan et al. (2018) did not share parts of the code implementing semantic segmentation.[4] These circumstances make reporting the accuracy of the implementation-specific baselines indispensable, which has thus become the standard practice in more recent previous (Chen et al., 2020; 2021) and related works (Gulrajani and Lopez-Paz, 2020).

## B    IABN

**Selecting $\alpha$.**    Fig. 4 shows a detailed plot of the influence of $\alpha$ on the segmentation accuracy, both on the development set of WildDash and on the target domains. We observe that the maximum accuracy on the development set is attained with $\alpha = 0.1$. Clearly, there is no guarantee that value $0.1$ is the optimal one for the target domains. However, choosing $\alpha$ based on the development set is in line with the established practice in machine learning: Tuning model hyperparameters is not allowed on the test sets (*i. e.* Cityscapes, BDD, IDD), but is only possible on the validation set (WildDash). In general, the hyperparameters found to be optimal on the validation set are not guaranteed to remain so on the test set. Nevertheless, our empirical results show consistent improvements over the baselines across all scenarios, despite $\alpha$ having been picked based on the validation dataset.

**SYNTHIA as the training set.**    Due to space constraints, we limited our study of IaBN in the main paper to the scenario of using GTA as the source data (*cf.* Table 1). Here, we extend this study by training our models on SYNTHIA instead. Table 5 reports the segmentation accuracy (in terms of IoU) and the expected calibration error (ECE) for this case. Notably, IaBN can still provide benefits for the expected segmentation accuracy if *p*-BN (*i. e.* using target instance normalization statistics) fails to improve over the *t*-BN baseline, as is the case with ResNet-50 in Table 5; the results remain on par with the *t*-BN baseline even when *p*-BN is significantly worse than *t*-BN. In regard to calibration quality, the results are consistent with our model trained on GTA (*cf.* Table 1): Not only does IaBN improve prediction calibration of the baseline, it again exhibits a complementary effect with MC-Dropout. Overall, the combined results from Tables 1 and 5 demonstrate that IaBN improves both the model accuracy and the calibration quality of the predictions in the out-of-distribution setting irrespective of the backbone network and specifics of the source data.

---

[2]We use Pillow library (https://pillow.readthedocs.io) to implement photometric augmentation.
[3]https://github.com/xyyue/DRPC/issues
[4]https://github.com/XingangPan/IBN-Net

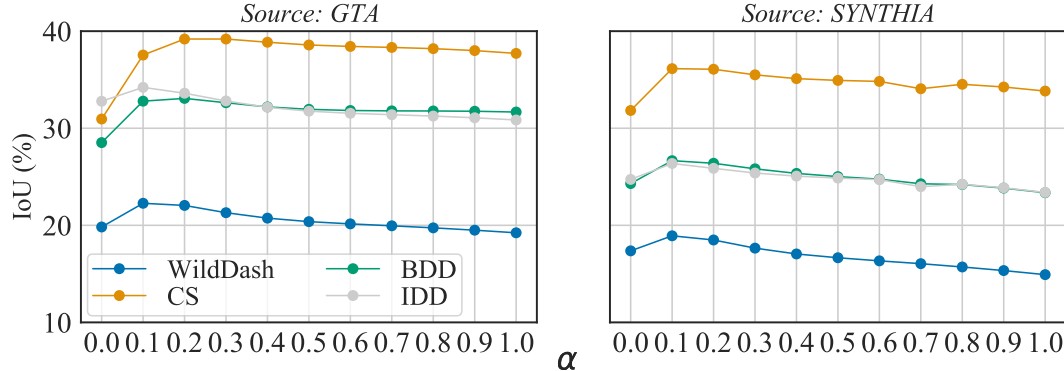

Figure 4: *Mean IoU (%, ↑) using IaBN based on the optimal alpha on the development set (WildDash).* We report scores for the target domains (Cityscapes, BDD, IDD) for the ResNet-50 backbone after training on GTA *(left)* and SYNTHIA *(right)*.

Table 5: *(a) Segmentation accuracy using IaBN.* We report the mean IoU (%) on three target domains (Cityscapes, BDD, IDD) across both backbones. As before, *t*-BN denotes train BN (Ioffe and Szegedy, 2015), while *p*-BN refers to prediction-time BN (Nado et al., 2020). *(b) ECE (%) for IaBN and MC-Dropout (Gal and Ghahramani, 2016).* We report scores for three target domains (Cityscapes, BDD, IDD) across both backbones. We trained the networks on SYNTHIA in both cases.

<table>
<tr><td colspan="5" align="center">(a)</td><td colspan="4" align="center">(b)</td></tr>
<tr><td rowspan="2">Method</td><td colspan="4" align="center">*IoU (%, ↑)*</td><td rowspan="2">Method</td><td colspan="3" align="center">*ECE (%, ↓)*</td></tr>
<tr><td>CS</td><td>BDD</td><td>IDD</td><td>Mean</td><td>CS</td><td>BDD</td><td>IDD</td></tr>
<tr><td>ResNet-50</td><td></td><td></td><td></td><td></td><td>ResNet-50 (Baseline)</td><td>37.50</td><td>43.19</td><td>40.11</td></tr>
<tr><td>w/ *t*-BN</td><td>31.83</td><td>24.30</td><td>24.73</td><td>26.95</td><td>w/ IaBN</td><td>30.96</td><td>33.27</td><td>36.31</td></tr>
<tr><td>w/ *p*-BN</td><td>33.83</td><td>23.36</td><td>23.39</td><td>26.86</td><td>w/ MC-Dropout</td><td>34.82</td><td>37.30</td><td>36.63</td></tr>
<tr><td>w/ IaBN *(ours)*</td><td>**36.14**</td><td>**26.66**</td><td>**26.37**</td><td>**29.72**</td><td>w/ MC-Dropout + IaBN</td><td>**30.66**</td><td>**33.06**</td><td>**35.60**</td></tr>
<tr><td>ResNet-101</td><td></td><td></td><td></td><td></td><td>ResNet-101 (Baseline)</td><td>31.39</td><td>33.77</td><td>36.56</td></tr>
<tr><td>w/ *t*-BN</td><td>37.25</td><td>**29.32**</td><td>27.19</td><td>31.25</td><td>w/ IaBN</td><td>30.33</td><td>31.83</td><td>36.26</td></tr>
<tr><td>w/ *p*-BN</td><td>34.58</td><td>24.24</td><td>22.32</td><td>27.05</td><td>w/ MC-Dropout</td><td>32.73</td><td>32.76</td><td>34.07</td></tr>
<tr><td>w/ IaBN *(ours)*</td><td>**38.01**</td><td>28.66</td><td>**27.28**</td><td>**31.32**</td><td>w/ MC-Dropout + IaBN</td><td>**27.71**</td><td>**30.48**</td><td>**32.67**</td></tr>
</table>

Table 6: *The role of the augmentation type in Seg-TTT.* We report mean IoU (%, ↑) and runtime (ms, ↓) for TTA (Simonyan and Zisserman, 2015) and our Seg-TTT for the GTA source domain and the Cityscapes target domain for the ResNet-50 backbone.

| Method | TTA | | Seg-TTT | |
| --- | --- | --- | --- | --- |
| | IoU | Runtime | IoU | Runtime |
| Baseline | 30.95 | **314** | 31.47 | **7135** |
| IaBN | 37.54 | **314** | 39.04 | **7135** |
| Multiple scales | 42.27 | 749 | 44.92 | 7272 |
| Horizontal flipping | 38.01 | 986 | 39.33 | 7593 |
| Grayscaling | 37.96 | 908 | 39.65 | 7193 |
| Multiple scales + horizontal flipping | 42.48 | 1316 | 44.94 | 7890 |
| Multiple scales + grayscaling | 42.28 | 1202 | 45.06 | 7486 |
| Multiple scales + horizontal flipping + grayscaling | **42.56** | 1308 | **45.13** | 7862 |

## C  SEG-TTT

**Further implementation details.** For Seg-TTT, we only update the model parameters in the layers conv4_x, conv5_x, and the classification head for the ResNet-50 backbone. Due to the higher computational cost of the ResNet-101 backbone, we only train the layers conv5_x and the classification head in this case.

Table 7: *Runtime (ms) with TTA (Simonyan and Zisserman, 2015) or Seg-TTT.* We report the runtime for both ResNet-50 and ResNet-101 on three dominant resolutions of $2048 \times 1024$, $1280 \times 720$, and $1920 \times 1080$, corresponding to the target domains Cityscapes, BDD, and IDD, respectively.

| Method | Input resolution | | | |
| --- | --- | --- | --- | --- |
| | CS | BDD | IDD | Mean |
| ResNet-50 (single-scale) | **314** | **136** | **214** | **221** |
| w/ TTA | 1308 | 713 | 766 | 929 |
| w/ Seg-TTT *(ours)* | 7862 | 3742 | 5307 | 5637 |
| ResNet-101 (single-scale) | **458** | **239** | **252** | **316** |
| w/ TTA | 1519 | 766 | 860 | 1048 |
| w/ Seg-TTT *(ours)* | 9060 | 4241 | 6142 | 6481 |

**The choice of augmentation strategies.** We verify the influence of the augmentation type used by Seg-TTT for test-time training. Recall from Sec. 5.2 that we use multiple scales with horizontal flipping and grayscaling to augment one image sample. We compare a flipping-only, scaling-only, and grayscaling-only version of our Seg-TTT to the combination of flipping, grayscaling, and scaling, which we used in the main text. We used a ResNet-50 backbone trained on GTA and report the test-time accuracy on Cityscapes in Table 6. We observe a significant boost in accuracy in comparison to our IaBN baseline with no augmentations. Furthermore, we show that using multiple scales is more important than flipping for Seg-TTT. Note that the augmentations used do not impact runtime in a significant way, since the batch sizes between these setups vary insignificantly; it is the backpropagation that dominates the main computational footprint. Varying the number of iterations, as studied in Sec. 5.3, provides a more flexible mechanism for accuracy-runtime trade-off.

**Inference time.** Table 7 compares the inference time of our Seg-TTT *w. r. t.* test-time augmentation (TTA) and single-scale inference across a range of input resolutions available in the target domains. We obtain these results by running inference on a single NVIDIA GeForce RTX 2080 GPU. To improve the runtime estimate, for each dataset we average the inference time over the complete image set. This is to account for small deviations in the input resolution (*e. g.*, IDD mostly contains images of resolution $1920 \times 1080$, but also has images with resolution $1280 \times 720$). Since our Seg-TTT uses 10 update iterations, the increase in the inference time *w. r. t.* TTA is expected. Although such cost may be detrimental for real-time applications, the clear accuracy benefits of Seg-TTT (*cf.* Table 4) may potentially appeal to use cases where the importance of the prediction quality outweighs the overhead in the frame rate. Furthermore, we investigated the influence of using the automatic mixed precision module in PyTorch and its influence on the inference runtime. While maintaining an identical IoU on Cityscapes using the ResNet-50 backbone, mixed precision achieves a runtime of 5746 ms compared to 7862 ms using single precision using $N_t = 10$ for our Seg-TTT. With $N_t = 3$ we are even able to decrease the runtime of Seg-TTT to 2914 ms.

We additionally provide runtime-accuracy plots for GTA $\to$ BDD and GTA $\to$ IDD generalization in Fig. 5. The data supports our conclusions drawn on GTA $\to$ Cityscapes generalization (*cf.* Sec. 5.3) that *(i)* Seg-TTT provides clear advantages in segmentation accuracy over baselines at a reasonable increase of the inference time; *(ii)* it is both more accurate and more efficient than model ensembles; and *(iii)* it exhibits a flexible runtime-accuracy trade-off by means of varying the number of update iterations and the number of the layers to adjust.

## D    TESTING IaBN AND SEG-TTT WITH OTHER ARCHITECTURES

Our approach generalizes to more recent architectures. We trained four state-of-the-art segmentation models on GTA: DeepLabv3+ (Chen et al., 2018b) with both a ResNet-50 and ResNet-101 backbone as well as HRNet-W18 and HRNet-W48 (Wang et al., 2021b). Table 8 reports consistent and substantial improvement of the mean IoU over the baseline, across all these architectures and the target domains.

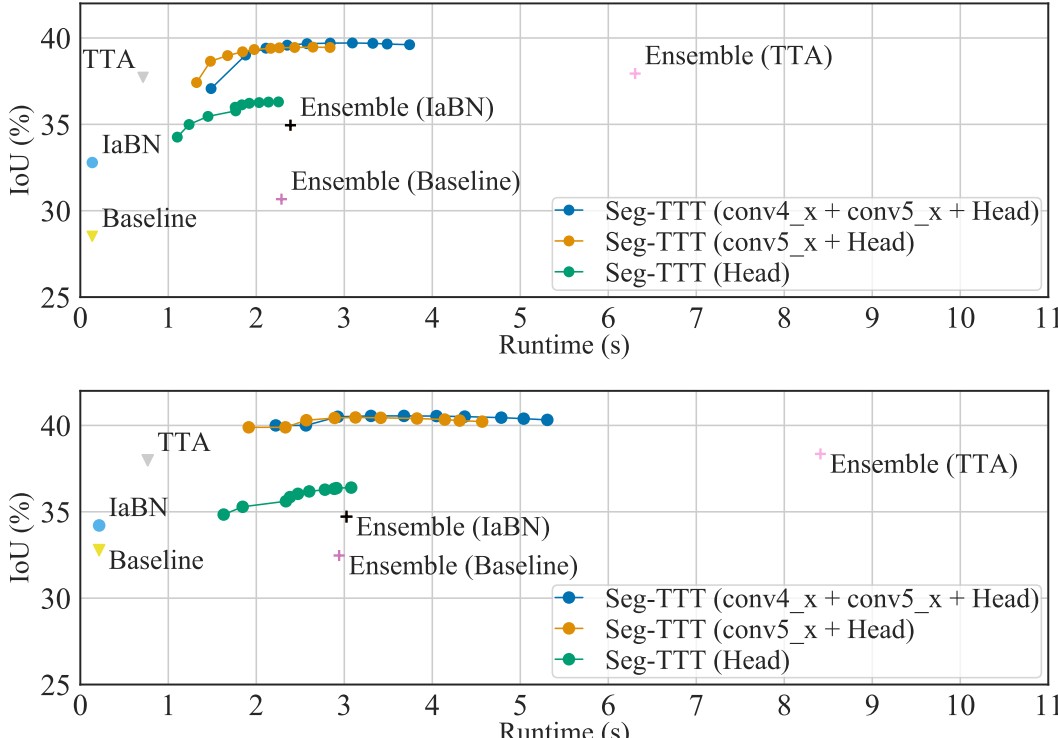

Figure 5: *Runtime-accuracy comparison on GTA → BDD (top) and GTA → IDD (bottom) generalization.* The curves on this plot trace Seg-TTT iterations, *i. e.* the first point corresponds to $N_t = 1$, while the last shows $N_t = 10$. While Seg-TTT increases the inference time of the baseline and TTA for the sake of improved accuracy, it is still more efficient and accurate than model ensembles of 10 networks. The choice of the layers for Seg-TTT updates (the naming follows He et al. (2016)) further provides a favorable runtime-accuracy trade-off. Runtimes are computed on a single NVIDIA GeForce RTX 2080 GPU.

Table 8: *Mean IoU (%) using our IaBN + Seg-TTT integrated with DeepLabv3+ (Chen et al., 2018b) based on a ResNet-50 and ResNet-101 backbone as well as with HRNet-W18 and HRNet-W48 (Wang et al., 2021b).* We observe substantial improvements of the segmentation accuracy on all three target domains (Cityscapes, BDD, and IDD) after training on GTA.

| Method | Target domains | | | |
| --- | --- | --- | --- | --- |
| | CS | BDD | IDD | Mean |
| DeepLabv3+ ResNet-50 (Baseline) | 37.51 | 35.45 | 37.50 | 36.82 |
| w/ IaBN + Seg-TTT *(ours)* | **46.56** | **43.17** | **44.07** | **44.60** |
| DeepLabv3+ ResNet-101 (Baseline) | 38.19 | 37.05 | 38.22 | 37.82 |
| w/ IaBN + Seg-TTT *(ours)* | **48.14** | **44.52** | **45.72** | **46.13** |
| HRNet-W18 (Baseline) | 33.08 | 29.40 | 32.97 | 31.82 |
| w/ IaBN + Seg-TTT *(ours)* | **44.05** | **38.29** | **43.78** | **42.04** |
| HRNet-W48 (Baseline) | 34.66 | 30.85 | 34.64 | 33.38 |
| w/ IaBN + Seg-TTT *(ours)* | **48.82** | **42.79** | **43.74** | **45.12** |

# E    COMPARISON TO CONTEMPORANEOUS WORK

In Table 9, we compare our method to the more recent approaches: RobustNet (Choi et al., 2021) and FSDR (Huang et al., 2021). Choi et al. (2021) disentangle domain-specific and domain-invariant properties from higher-order statistics of the feature representation by using an instance selective

Table 9: *Mean IoU (%) comparison to state-of-the-art domain generalization methods* for both source domains (GTA, SYNTHIA) as well as three target domains (Cityscapes, Mapillary, BDD). We compare to RobustNet (Choi et al., 2021) and FSDR (Huang et al., 2021). ($\ddagger$) and ($\dagger\dagger$) denote the use of FCN (Long et al., 2015) and DeepLabv3+ (Chen et al., 2018b) architectures, respectively.

| | Method | Backbone: ResNet-50 | | | Backbone: ResNet-101 | | |
|---|---|---|---|---|---|---|---|
| | | CS | Mapillary | BDD | CS | Mapillary | BDD |
| *GTA* | No Adapt RobustNet†† | 28.95 / 36.58 ↑7.63 | 28.18 / 40.33 ↑12.15 | 25.14 / 35.20 ↑10.06 | – | – | – |
| *GTA* | No Adapt Ours†† | 37.75 / **46.11** ↑8.36 | 40.36 / **49.53** ↑9.17 | 33.70 / **41.75** ↑8.05 | – | – | – |
| *GTA* | No Adapt FSDR‡ | – | – | – | 33.4 / 44.8 ↑11.4 | 27.9 / 43.4 ↑15.5 | 27.3 / **41.2** ↑13.9 |
| *GTA* | No Adapt Ours | 30.95 / **45.13** ↑14.18 | 34.56 / **47.49** ↑12.93 | 28.52 / **39.61** ↑11.09 | 32.90 / **46.99** ↑14.09 | 36.00 / **47.49** ↑11.49 | 32.54 / 40.21 ↑7.67 |
| *SYNTHIA* | No Adapt FSDR‡ | – | – | – | - / 40.8 | - / 39.6 | - / **37.4** |
| *SYNTHIA* | No Adapt Ours | 31.83 / **41.60** ↑9.77 | 33.41 / **41.21** ↑7.80 | 24.30 / **33.35** ↑9.05 | 37.25 / **42.32** ↑5.07 | 36.84 / **41.20** ↑4.36 | 29.32 / 33.27 ↑3.95 |

whitening loss. FSDR (Huang et al., 2021) randomizes images in the frequency space by keeping domain-invariant frequency components and varying the domain-sensitive frequency components.

For a fair comparison with RobustNet, we trained a DeepLabv3+ model with the output stride of 16 in contrast to the output stride of 8 used by default (*e. g.*, in Table 8). We also remark that FSDR used the target domains for hyperparameter tuning and model selection, which gives it a clear advantage.[5] Nonetheless, we outperform FSDR in terms of IoU on two out of three target domains using ResNet-101 backbone. Remarkably, our baseline training schedule and data augmentation alone already reach the accuracy of RobustNet without any additional regularization or changes to the model architecture. Our IaBN with Seg-TTT further improve over these results by significant margins (more than 8% IoU) across all tested target domains.

## F   DATASET DETAILS

**GTA.**   GTA (Richter et al., 2016) is a street view dataset generated semi-automatically from the computer game Grand Theft Auto V. The dataset consists of 12,403 training images, 6,382 validation images, and 6,181 testing images of resolution 1914 × 1052 with 19 different semantic classes.

**SYNTHIA.**   We use the SYNTHIA-RAND-CITYSCAPES subset of the synthetic dataset SYNTHIA (Ros et al., 2016), which contains 9,400 images, and has 16 semantic classes in common with GTA. Images have a resolution of 1280 × 760 pixels.

**WildDash.**   This dataset was developed to evaluate models regarding their robustness for driving scenarios under real-world conditions. It comprises 4256 images of real-world scenes with a resolution of 1920 × 1080 pixels.

**Cityscapes.**   Cityscapes (Cordts et al., 2016) is an ego-centric street-scene dataset and contains 5,000 high-resolution images with 2048 × 1024 pixels. It is split into 2,975 train, 500 val, and 1,525 test images with 19 semantic classes being annotated.

**BDD.**   BDD (Yu et al., 2020) is a driving video dataset, which also contains semantic labelings with the identical 19 classes as in the other datasets. Images have a resolution of 1280 × 720 pixels. The training, validation, and test sets contain 7,000, 1,000, and 2,000 images, respectively.

**IDD.**   IDD (Varma et al., 2019) is a dataset for road scene understanding in unstructured environments. It contains 10,003 images annotated with 34 classes even though we only evaluate on the 19 classes

---

[5]https://github.com/jxhuang0508/FSDR/issues/2#issuecomment-910089417

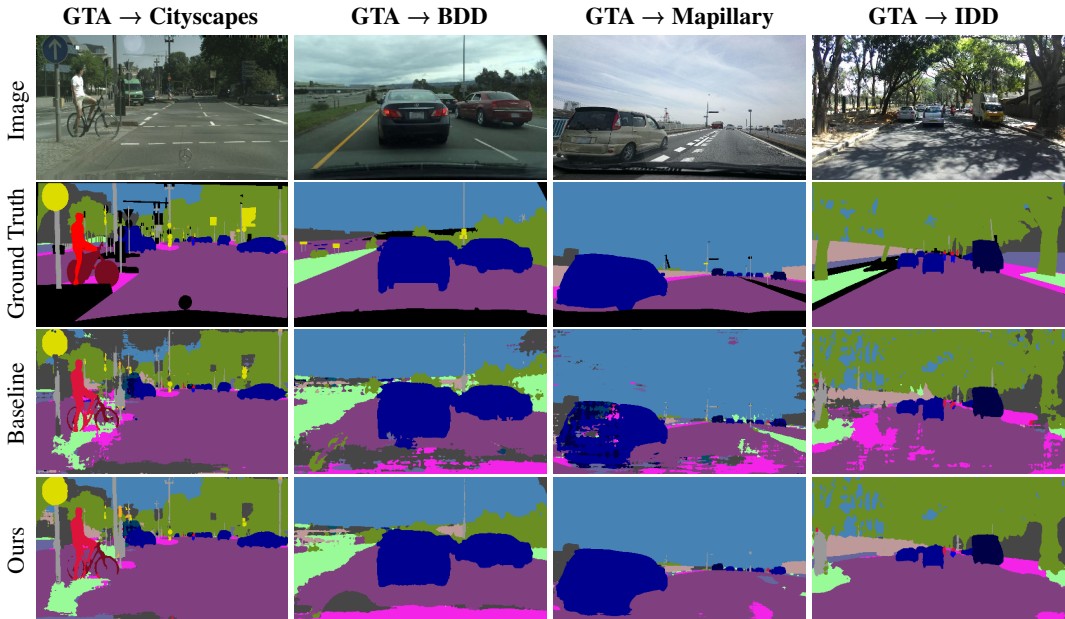

Figure 6: *Qualitative semantic segmentation results for generalization from GTA to Cityscapes, BDD, Mapillary, and IDD for the ResNet-101 backbone.*

overlapping with the other datasets. IDD is split into 6,993 training images, 981 validation images, and 2,029 test images.

**Mapillary.** Annotations from Mapillary (Neuhold et al., 2017) contain 66 object classes; analogously to IDD we only evaluate on the 19 classes overlapping with the other datasets. The dataset is split into a training set with 18,000 images and a validation set with 2,000 images with a minimum resolution of 1920 × 1080 pixels.

## G QUALITATIVE EXAMPLES

We provide additional qualitative results by running inference on three models: ResNet-101 trained on GTA in Fig. 6; ResNet-50 and ResNet-101 trained on SYNTHIA in Figs. 7 and 8, respectively. Similar to our observations in Sec. 6.3 (*cf.* main text), our approach exhibits more homogenous semantic masks with visibly fewer jagged-shaped artifacts than the baseline (*e. g.*, "sidewalk" false positives in Fig. 6). Models with our IaBN and Seg-TTT may still struggle with cases of mislabeling regions with an incorrect, but semantically related class. For example, the model often assigns "sidewalk" to the road pixels from BDD and IDD in Figs. 7 and 8. This is an expected outcome if the erroneous labels are already contained in the pseudo labels of the initial prediction, on which Seg-TTT relies. These failure cases occur more frequently if the domain shift between the train and the test distributions is more significant, such as between SYNTHIA and BDD, which can lead to poorly calibrated predictions. Since applying IaBN results in improved calibration (*cf.* Table 1), it alleviates this issue, and Seg-TTT can cope well with milder domain shift scenarios, as a result. As examples on Cityscapes and Mapillary in Figs. 7 and 8 show, despite the baseline model exhibiting some degree of this failure mode, our inference method visibly rectifies these errors.

We additionally ran our inference on video sequences and include the results as part of our supplementary material. Confirming our previous analysis of the qualitative results (*cf.* Sec. 5.3), we observe that our approach clearly improves the segmentation quality and removes some of the most pathological failure modes of the baseline (*e. g.*, the lower middle part of the frame area).

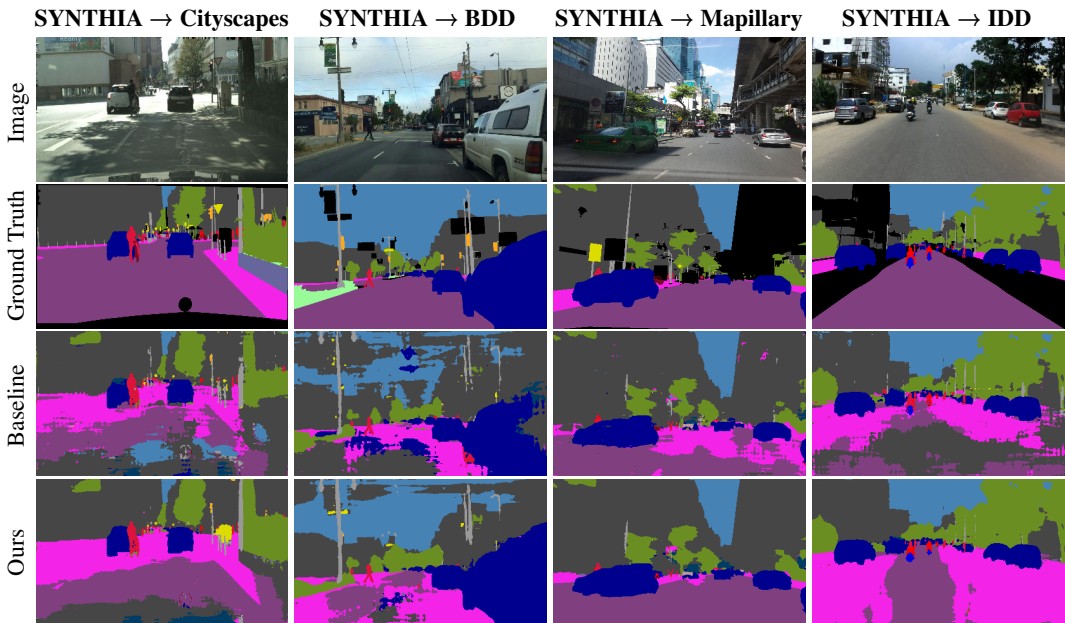

Figure 7: *Qualitative semantic segmentation results for generalization from SYNTHIA to Cityscapes, BDD, Mapillary, and IDD for the ResNet-50 backbone.*

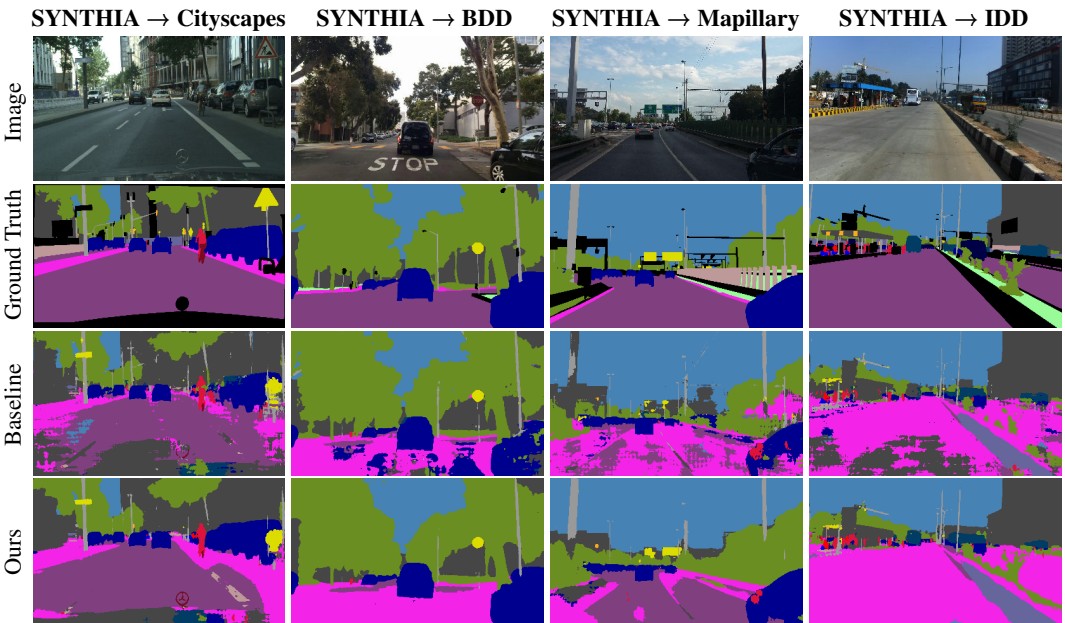

Figure 8: *Qualitative semantic segmentation results for generalization from SYNTHIA to Cityscapes, BDD, Mapillary, and IDD for the ResNet-101 backbone.*

## H  FAILURE CASES

Conceptually, if the initial semantic prediction is incorrect and confident (due to miscalibration), it is likely to end up in the pseudo labels and lead astray the adaptation process. The rightmost columns in Fig. 7 and Fig. 8 already visualized this failure mode: the baseline largely misclassifies "road" as the "sidewalk" class. We have further extended these examples with Fig. 9. Misguided by incorrect pseudo labels, our test-time adaptation may exacerbate this issue, *i. e.* propagate the incorrect label to the areas sharing the same appearance. To gain further insights, we investigated this issue statistically.

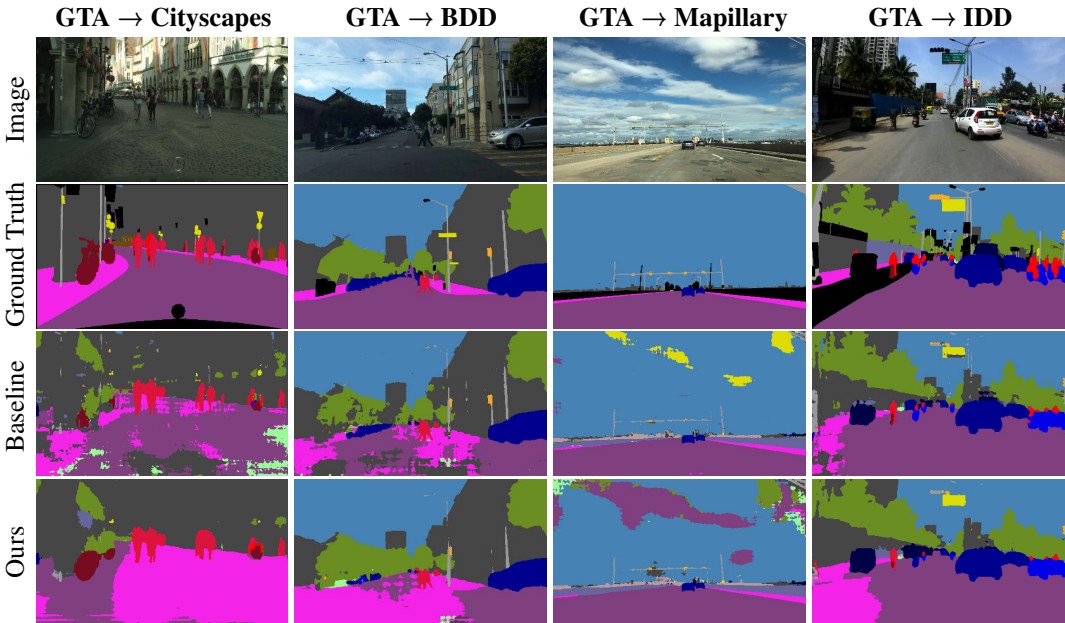

Figure 9: *Failure cases of semantic segmentation for generalization results from GTA to Cityscapes, BDD, Mapillary, and IDD* for the ResNet-50 backbone.

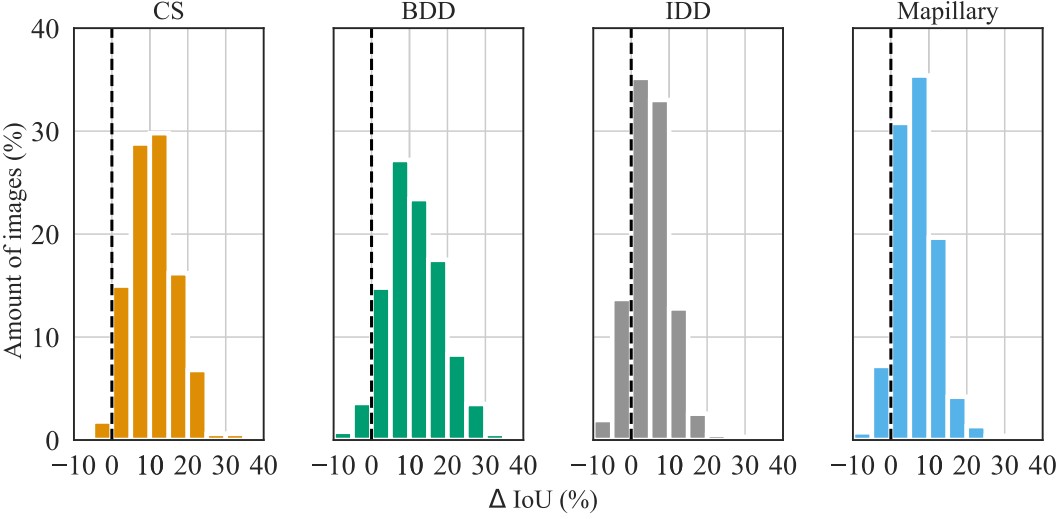

Figure 10: *Density distribution of the individual images for generalization from source domain (GTA) to target domains (Cityscapes, BDD, IDD and Mapillary)* for the ResNet-50 backbone. We visualize the relative improvement of our test-time inference strategy w.r.t. the baseline in terms of accuracy with $\Delta$ IoU (%).

The histograms in Fig. 10 show relative improvement of our test-time inference strategy *w. r. t.* the baseline in terms of IoU on four validation sets. We conclude that such decrease in segmentation accuracy is actually rather rare: less than 10% of the images, on average, exhibit lower accuracy. In most such cases the accuracy reduces only marginally (by less than 5%), and only a fraction of images (less than 1% on average) deteriorate in accuracy by at most 10%. The overwhelming majority of the image samples benefit from our test-time adaptive process and increase in accuracy by up to 35% IoU.

## I  CODE

We provide an implementation of our approach as part of the supplementary material. The enclosed `README` provides further details on running this code for both training and inference. Upon acceptance, this code will be released to the community under the Apache License. Furthermore, to reproduce the state-of-the-art results of our models reported in Table 4, we make their snapshots available via an anonymous link provided in the `README` file.

