# OpenReview forum: "Adaptive Generalization for Semantic Segmentation"
_ICLR.cc/2022/Conference — ICLR 2022 Submitted_

### Official Review · Reviewer_FDdf · 2021-10-31

**Correctness:** 4
**Technical Novelty And Significance:** 2
**Empirical Novelty And Significance:** 2
**Recommendation:** 5
**Confidence:** 4

**Details Of Ethics Concerns:**

I don't have any ethics concerns.

**Main Review:**

There are strengths of this paper:
+ The paper is overall well-written and the reviewer can easily follow the whole draft and get the key points of this paper.

+ This paper shows consistent better than previous method. Besides, this paper also analyze two components of this work to demonstrate the effectiveness of their work. According to the experiments  and ablation study using GTA5, Cityscapes, BDD100k, IDD, their method achieves more favorable results than some previous methods including t-BN and p-BN, where their method is a tradeoff between them.  Besides, this paper also shows better results than previous test-time training method.

The weakness of this paper are as follows:
- The novelty of this work is the main drawback. The first point instance-adaptive normalization is a trivial solution, which is the tradeoff of previous t-BN and p-BN. Second, there are a lot of work applying pseudo labels in semantic segmentation, such as Naive-students [ECCV2020]. Even though this work leverages pseudo labels for a different application, the technique contribution is not strong.

- Comparison to more previous work [a,b,c] in domain generalization in semantic segmentation is needed.
[a] RobustNet: Improving Domain Generalization in Urban-Scene Segmentation via Instance Selective Whitening, CVPR, 2021.
[b] FSDR: Frequency Space Domain Randomization for Domain Generalization, CVPR, 2021.
[c] Domain Randomization and Pyramid Consistency: Simulation-to-Real Generalization without Accessing Target Domain Data, ICCV, 2019.

- In instance-adaptive batch normalization paragraph, the authors use \head{u}_{c}^{s}, but in Eq (4), the authors use u_{c}^{s}. Need to confirm the notations in the paragraph.

- In the experiments, the paper uses 0.1, 0 and 1 for the alpha in Eq 4, where 0 and 1 are the baseline method. Is the method sensitive to alpha in acheving good results, more analysis is needed. For various target datasets, they have different levels of domain gap, it is unclear how to choose the parameter for \alpha.


Question:
1. What is the motivation to test on the expected calibration error with a combination  of MCDropout ?
2. How to choose alpha for Eq. 4?




**Summary Of The Paper:**

This paper studies an existing problem - domain generalization for semantic segmentation of urban scenes. The technique keys of this work has two main points: instance-adaptive batch normalization and testing-time training using pseudo labels. Instance-adaptive batch normalization aims to bias to the data distribution of individual testing examples, which is a tradeoff of previous t-BN and p-BN. This paper generate pseudo labels for test-time training, which is not quite novel. To evaluate the effectiveness of the proposed method, this paper conducts experiments on GTA5/SYNTHIA -> Cityscapes, BDD100k and IDD, and shows better results.


**Summary Of The Review:**

This paper presents a new model and pipeline for domain generalization in semantic segmentation. Although this work shows better results than previous work. Lacking of novelty and insights is the main drawback of this work. Besides, the experiments can be improved, that more comparisons with previous work is helpful. Therefore, the reviewer rates this paper marginally below the acceptance threshold.

---

> ### Author Response · Authors · 2021-11-16
> **Response to reviewer FDdf**
>
> Thank you for your careful review and pointing out some relevant work.
>
> > Novelty.
>
> We kindly refer to the general comment for our response regarding novelty.
> Although we employ familiar technical components, we believe our study to bear sufficient *empirical novelty*: the particular implementation of the test-time adaptation process we study has not been investigated as comprehensively before.
>
> > Comparison to more related work [a,b].
>
> Thank you for pointing out these contemporaneous (as per ICLR 2022 guidelines) works.
> We have included this comparison in Appendix E, Tab. 9.
> For a fair comparison with RobustNet (Choi et al., 2021), we trained a DeepLabv3+ model with the output stride of 16.
> Remarkably, our baseline training schedule and data augmentation (cf. Appendix A) alone already reach the accuracy of RobustNet without any additional regularization or changes to the model architecture.
> Our IaBN with Seg-TTT further improves over these results by significant margins (more than 8% IoU on average) across all tested target domains.
> We also remark that FSDR (Huang et al., 2021) used the target domains for hyperparameter tuning and model selection, which gives it a clear advantage (see https://github.com/jxhuang0508/FSDR/issues/2#issuecomment-910089417).
> However, we still outperform FSDR in terms of segmentation accuracy on two out of three target domains using a ResNet-101 backbone.
>
> > Comparison to [c] (Yue et al., 2019).
>
> We have actually provided this comparison in Tab. 4, which shows significant improvement over the approach by Yue et al. (2019) across virtually all tested scenarios (the only exception is SYNTHIA $\rightarrow$ BDD with a ResNet-101 backbone, where the difference is marginal).
>
> > Notation in Eq. (4).
>
> Thank you. We have corrected the notation in Eq. (4) of the running statistics for the source domain.
>
> > How to choose $\alpha$? Is the method sensitive to $\alpha$?
>
> Since we define $\alpha$ as the model hyperparameter, the established protocol in machine learning is to use a validation set (or, a development set) for its finetuning.
> In this work, we used WildDash as the validation set (cf. Sec. 5).
>
> We extended Fig. 4 to include the accuracy for intermediate values of $\alpha$ across the target domains.
> The sensitivity of our adaptation scheme to $\alpha$ depends on the choice of the source domain and the target domain.
> For example, trained on GTA our adaptation process on Cityscapes is rather robust to the choice of $\alpha$ as long as it is larger than or equal to 0.1.
> However, if the model is trained on SYNTHIA, a suboptimal choice of $\alpha$ may decrease the model accuracy.
> This underscores the need for a validation set for selecting $\alpha$.
> Using one of the target domains would invalidate our evaluation protocol for out-of-distribution robustness, hence we use an independent validation set.
> Note that even though the optimal alpha on the validation set WildDash may not coincide with its optimal value on the target domains, it nevertheless leads to a new state-of-the-art in generalization accuracy for semantic segmentation.
>
> > Motivation for ECE with MC-Dropout.
>
> MC-Dropout is a widely adopted technique for improving model calibration.
> Although we do not use MC-Dropout in our adaptation process, we believe that the complementary effect of IaBN on MC-Dropout (both in calibration and accuracy) is rather surprising and may be of interest to broader research circles.

---

> > ### Comment · Reviewer_FDdf · 2021-11-30
> > **Still believe novelty is not strong enough to accept**
> >
> > Thanks a lot for preparing the rebuttal, it resolves some concerns. However, I still believe the novelty of this work is the most critical point not to accept, after reading all the reviews that reviewers tfFH and YkXo also point out the same point.

---

> > > ### Author Response · Authors · 2021-11-30
> > > **Response to reviewer FDdf**
> > >
> > > Thank you for your feedback!\
> > > We closely inspected all the previous works referenced in the reviews, but could not find any meaningful overlap with the *empirical* novelty of our study. We believe that our analysis offers important insights and a fresh perspective on domain generalization for semantic segmentation: we are not aware of any prior work that demonstrates significant accuracy benefits from adjusting the *inference* process (which we do), rather than the *training* scheme, as done in previous work.
> > >
> > > We would be very grateful if the reviewer could point out any concrete references to previous work that propose a comparable approach as ours for (test-time) domain generalization.

---

### Official Review · Reviewer_YkXo · 2021-11-02

**Correctness:** 3
**Technical Novelty And Significance:** 3
**Empirical Novelty And Significance:** 3
**Recommendation:** 5
**Confidence:** 3

**Main Review:**

Pros:
1.	The writing is good, making the paper very easy to understand.
2.	Both the two modules are conceptually simple.
3.	Extensive experiments have been performed for evaluation.
4.	Code is provided.

Cons:
1.	My major concern goes to the novelty of the paper. The proposed normalization technique is incremental. Instance-level normalization is a smart way for test-time adaptation, which has been well studied [1][2][3]. For the test-time training part, it is also hard to claim a unique contribution since it mostly follows a pseudo labeling-based segmentation paradigm.

  [1] Test-time adaptable neural networks for robust medical image segmentation
  [2] Autoencoder based self-supervised test-time adaptation for medical image analysis
  [3] Tent: Fully Test-Time Adaptation by Entropy Minimization

2.	It is not clear to me the advantage to learn \alpha using a validation set, rather than online learning during test-time finetuning.
3.	The necessity to use an independent development set for model/parameter selection needs more studies. It is also not clear to me how to select a universe development set that can fit any target domain, or should we find a unique dev set for each target domain.
4.	The model is currently evaluated using relatively lightweight backbones (Res50, HRNet18), and it is not clear whether the method can work well for larger models (e.g., Res101, HRNet48) and lead to consistent improvements.


**Summary Of The Paper:**

The article proposes two techniques for test-time adaptation. The first is instance-adaptive bn, to combine statistics from source data with each single test sample.  The second is test-time training, based on pseudo labeling of the test sample. The two parts lead to promising results.

**Summary Of The Review:**

Overall, the paper has some merits (e.g., good writing, promising results). However, the technical contributions seems to be minor. I am concerning that the paper may be still under the level of ICLR.

---

> ### Author Response · Authors · 2021-11-16
> **Response to reviewer YkXo**
>
> Thank you for your valuable feedback and interesting suggestions.
>
> > Novelty.
>
> We kindly refer to the general comment above for our response regarding novelty.
>
> > Novelty w.r.t. [1,2].
>
> Thank you for referencing these relevant works.
> Although both of these works deploy test-time training, the specific implementation of the idea is conceptually different from IaBN and Seg-TTT and more computationally involved.
> In more detail, both works employ a test-time reconstruction objective, which relies on a standalone auto-encoder, instead of simple pseudo-labels constructed in our work.
> This not only introduces additional training complexity, but also increases the test runtime: [2] needs at least 15 seconds for adaptation, [1] requires about 1 hour.
> The adaptation process we study is substantially more efficient (350ms - 6s) and further offers a compelling runtime-accuracy trade-off (cf. Fig. 2).
>
> > Novelty w.r.t. [3].
>
> We compare and discuss this work in a dedicated paragraph at the end of Sec. 5.
> Tent, in fact, does not employ instance normalization, as does IaBN in our work.
> Instead, Tent minimizes the entropy of model predictions by updating only Batch Normalization parameters with gradient descent, as well as uses the running statistics of a *target batch*, rather than a single target sample.
> Tent is substantially less effective as we exemplify by training our model on GTA and testing it on Cityscapes: IaBN alone already outperforms Tent by 3.6% IoU; when complemented by Seg-TTT, it surpasses Tent by 7.7% IoU.
> Note that since Tent also requires 10 iterations of gradient descent, the computational footprint is comparable to the approach we study.
>
> > Why not learn $\alpha$ during test-time finetuning?
>
> Thank you for this interesting suggestion.
> We have actually experimented with learning an instance-specific $\alpha$ using our adaptation process.
> However, we did not find the accuracy benefits significant and consistent enough to justify the additional complexity this introduces to the optimization process, as opposed to our original strategy of simply maintaining $\alpha$ as a hyperparameter (i.e. a fixed value).
>
> > The necessity of an independent validation dataset. How to select a universe development set that can fit any target domain, or should we find a unique dev set for each target domain?
>
> As we study out-of-distribution model robustness, we cannot make strong assumptions about the test distribution, let alone use the test distribution itself for model training or hyperparameter tuning, as was done in previous work (e.g., Chen et al. (2021)).
> This motivates our use of a single validation set derived independently from the target domains.
> Without any knowledge about the target domains, we assume a uniform prior.
> In the context of street scenes, this implies that our development set should encompass an unbiased variety of geographic locations, weather conditions, as well as camera characteristics.
> We found WildDash to largely satisfy these constraints, hence adopt it as our development set.
> Sec. 5 provides further details.
>
> > Accuracy for stronger architectures.
>
> We kindly note that we tested DeepLabv1 both with ResNet-50 and ResNet-101 backbones.
> We have further extended Table 8 to include DeepLabv3+ with ResNet-101 and HRNet-W48, duplicated below for convenience.
> As expected, the results show consistent and substantial improvements across all target domains:
>
>  |Method|     CS     |     BDD    |      IDD    |       Mean   |
>  |-----------------|:--------:|:--------:|:--------:|:--------:|
>  |**DeepLabv3+ ResNet-101 (Baseline)**     | 38.19 | 37.05 | 38.22 | 37.82 |
>  |&nbsp; **w/ IaBN + Seg-TTT (ours)**     | **48.14** | **44.52** | **45.72** | **46.13** |
>  |**HRNet-W48 (Baseline)**     | 34.66 | 30.85 | 34.64 | 33.38 |
>  |&nbsp; **w/ IaBN + Seg-TTT (ours)**     | **48.82** | **42.79** | **43.74** | **45.12** |

---

### Official Review · Reviewer_Dswx · 2021-11-02

**Correctness:** 3
**Technical Novelty And Significance:** 3
**Empirical Novelty And Significance:** 3
**Recommendation:** 6
**Confidence:** 3

**Main Review:**

Strengths

- Test-time training is an interesting concept and the paper spans a nice logical bridge from the first, technical rather minor conribution, that improves calibration of the prediction to leverage areas that are likely predicted correctly to further improve th inference results.

- Interesting, but a bit heuristic, combination of test-time training and pseudo-labeling.

- Good results and broad evaluation.

Weaknesses:

- Inference speed: Test-time training is by definition rather costly and the proposed approach increases runtime by more than an order of magnitude when compared to the baseline. It is unclear how to improve the practicality of the approach.

- When comparing to Wang 2021a it is not clear to me how much improvement actually comes from using multi-scale predictions and augmentations. It would be great to have an experiment along these lines.

- No comparison to (offline) pseudo-labeling approaches (a recent example would be [A]). The overall strategy would also be applicable without test-time training in principle. I.e. one could derive pseudo-labels for the whole target dataset and adapt the classifier once on the full target dataset. While the assumption here is that only a single image from the target domain is available, the setting seems to be rather artificial as a few unlabeled images from the target domain should almost always be available (and test-time training might help on top of such a transfer procedure).

Other questions:

- It would be interesting to see some failure cases (if any). Does test-time training always improve results?

[A] Zou et al., PseudoSeg: Designing Pseudo Labels for Semantic Segmentation, ICLR 2021



**Summary Of The Paper:**

This paper contributes two techniques to improve generalization of semantic segmentation networks. The first technique is an adaptation of the test-time behaviour of batch normalization, where the statistics of the sample under consideration are blended into the training-time statistics of the batch-norm layer. The second technique is to use test time augmentations to first derive a pseudo-labeling from the sample and then to update parts of the weights to maximize the probability of the pseudo-labels. The paper additionally contributes a multi-dataset evaluation procedure and shows favorable performance of the proposed techniques on this evaluation protocol.

**Summary Of The Review:**

I think the overall approach is interesting, but ad-hoc in its design. It would be great to see some comparisons to existing pseudo-labeling approaches as I think that the setting with only a single image from the target domain artificial.

---

> ### Author Response · Authors · 2021-11-16
> **Response to reviewer Dswx**
>
> Thank you for your positive and thoughtful feedback.
>
> > It is unclear how to improve the practicality of the approach.
>
> We fully acknowledge that Seg-TTT provides accuracy benefits at some computational expense, which we thoroughly analyze for transparency (e.g., expanding on Wang et al. (2021a)).
> Improving the computational efficiency of gradient-based learning is the subject of a number of research domains, which are, however, outside the scope of this work.
> One example, is seeking faster convergence (i.e. using fewer iterations) with adaptive step sizes (e.g., as in Adam, AdaGrad, or RMSProp), higher-order optimization (e.g., LBFGS), or using implicit gradients [A].
> Low-precision computations [B], low-rank decomposition [C], or alternating update strategies [D] can offer further efficiency gains.
> With constantly evolving hardware, real-time test-time adaptive strategies may also become feasible even on mobile devices [E,F].
> We believe that the exciting results of our study provide strong incentives for future research efforts in these areas.
>
> [A] Rajeswaran et al., Meta-Learning with Implicit Gradients. In NeurIPS 2019.\
> [B] Wang et al., E2-Train: Training State-of-the-art CNNs with Over 80% Energy Savings. In NeurIPS 2019.\
> [C] Jaderberg et al., Speeding up Convolutional Neural Networks with Low Rank Expansions. In BMVC 2014.\
> [D] Tonioni et al., Real-time self-adaptive deep stereo. In CVPR 2019.\
> [E] Ha et al., Accelerating On-Device Learning with Layer-Wise Processor Selection Method on Unified Memory. In Sensors 2021.\
> [F] Mullapudi et al., Online Model Distillation for Efficient Video Inference. In ICCV 2019.
>
>
> > Influence of augmentations on Seg-TTT.
>
> We have revised Tab. 6 to include our baselines without the augmentations, and include a summary of the table below for convenience.
> We observe that using the augmentations at inference time, especially multiple scales, is crucial for obtaining high accuracy with Seg-TTT.
> Tent (Wang et al., 2021a), by contrast, does not use augmentations.
> The comparison to Tent in Sec. 5.3 suggests that test-time training by minimizing the entropy alone may be particularly prone to an amplification of errors in the original predictions of the model.
> This is remedied in our approach by improved model calibration and the confidence threshold allowing to generate high-precision pseudo labels.
>
>  | Method |     CS     |
>  |-----------------|:--------:|
>  |**Seg-TTT (Baseline)**     | 39.04 |
>  |**w/ multiple scales**     | 44.92 |
>  |**w/ horizontal flipping**     | 39.33 |
>  |**w/ grayscaling**     | 39.65 |
>  |**w/ multiple scales + horizontal flipping + grayscaling**     | **45.13** |
>
> > Comparison to offline pseudo-labeling; the setting seems to be rather artificial.
>
> We agree that a comparison to self-training approaches for semi-supervised learning and domain adaptation would be interesting.
> However, as we discuss at the end of Sec. 2, such scenarios are different from evaluating out-of-distribution robustness, which is our focus, hence are beyond the scope of this work.
> Our setup assumes no information transfer between the inference processes of different image samples, which is a widely adopted assumption for evaluation on test sets in the literature.
> This becomes a justified scenario in applications without real-time requirement (e.g., in satellite imaging), or when the frame rate is low (e.g., due to a low bandwidth of image transmission).
> Since domain adaptation and semi-supervised learning are also computationally involved and time-consuming procedures, adapting to a single sample may also present a more computationally appealing alternative (e.g., for edge devices).
>
> > Failure cases.
>
> Conceptually, if the initial semantic prediction is incorrect and confident (due to miscalibration), it is likely to end up in the pseudo labels and lead astray the adaptation process.
> The rightmost columns in Fig. 7 and 8 (cf. Appendix F) already visualized this failure mode: the baseline largely misclassifies "road" as the "sidewalk" class.
> We have further extended these examples with Fig. 9.
> Misguided by incorrect pseudo labels, our test-time adaptation may exacerbate this issue, i.e. propagate the incorrect label to the areas sharing the same appearance.
> To gain further insights, we investigated this issue statistically.
> The histograms in Fig. 10 show the relative improvement of our test-time inference strategy w.r.t. the baseline in terms of IoU on four validation sets.
> We conclude that such decrease in segmentation accuracy is actually rather rare: less than 10% of the images, on average, exhibit lower accuracy.
> In most such cases, the accuracy is reduced only marginally (by less than 5%), and only a fraction of images (less than 1% on average) deteriorate in accuracy by at most 10%.
> The overwhelming majority of the image samples benefits from our test-time adaptive process and increase in accuracy by up to 35% IoU.

---

### Official Review · Reviewer_tfFH · 2021-11-04

**Correctness:** 2
**Technical Novelty And Significance:** 2
**Empirical Novelty And Significance:** 2
**Recommendation:** 5
**Confidence:** 3

**Main Review:**

Pros)

The paper is well written and easy to understand. The experiments including comparison to baselines and ablation studies seem to be comprehensive.

Cons)

The idea on combining BN and IN has been proposed before. The novelty is limited. In addition, in this work, the hyperparameter, alpha, has to be decided by validation data.

TT-SEG is a simple method for test-time finetuning. Although the effectiveness was shown by the experiments, the novelty seems to be limited.

The description on baselines such as ASG and CSG in Related Work is too brief to understand the principle difference to the proposed method and among them.

Taking six seconds is too much from practical point of view. As the authors mentioned, the proposed method is unpractical for road scene analysis.

Comment)

Why was ECE used only for Table1(b) without showing MIoU ? Is MIoU also improved by introducing MC-Dropout ?


**Summary Of The Paper:**

This paper presents a method on generalization of segmentation from synthetic data to real street scene data. To adapt the model pre-trained with synthetic source domain data such as GTA and SYNTHIA to target domain such Cityscapes, BDD and IDD, the authors proposes Instance-adaptive Batch Normalization (IaBN). In addition, as a method to learn from a single sample, TT-SEG was proposed
in which a pseudo GT mask is estimated from augmented images with the initial model and the last parts of the network are finetuned with the pseudo GT mask. In the experiments, the proposed method which employs both IaBN and TT-SEG successfully outperformed all the baselines, which indicated the effectiveness of the proposed method.



**Summary Of The Review:**

The novelty of the proposed method is limited, since IaBN is an existing method and the method on one-sample adaptation is straight-forward. Although the effectiveness of the proposed method regarding performance was shown by the experiments, the required processing time was too much. From these points, this paper is regarded as being around the border.

---

> ### Author Response · Authors · 2021-11-16
> **Response to reviewer tfFH**
>
> Thank you for your valuable feedback.
>
> > Novelty.
>
> Please refer to our general response above regarding novelty.
> While we agree that BN and IN have been combined before, the specific implementation we use in our study is different from previous works.
> Since the reviewer seemed to disagree, we would be grateful if the reviewer could specify where their combination as in our IaBN (but also its composition with test-time training) has been published before.
> We are not aware of such work.
>
> > $\alpha$ has to be decided by validation data.
>
> We use a validation set to select our hyperparameters including $\alpha$, which is a well established practice in machine learning.
> This is a principled approach, and stands in contrast to previous and very recent works.
> For example, Chen et al. (2020), Chen et al. (2021) and Huang et al. (2021) use the *target* domain for validation, which effectively amounts to using the *test set* for hyperparameter tuning, since the same image set is eventually used to compare the model accuracy.
> Similar to our approach, Yue et al. (2019) also use an independent validation set.
> As we point out in Sec. 5, however, the choice of this validation set in (Yue et al., 2019) is different for each target domain, which results in *multiple* target-specific models.
> By contrast, our validation strategy allows to produce a *single* model with significantly improved accuracy (w.r.t. the baseline) on *all* target domains we tested.
> Nevertheless, we are happy to experiment with other model selection strategies, if advised by the reviewer.
>
> > Improved description of previous work (e.g. ASG, CSG).
>
> Thank you. We have revised the corresponding paragraph in Sec. 2 to make these distinctions clearer.
>
> While the focus in previous investigations was the training strategy (Chen et al., 2020) and model design (Pan et al., 2018), we exclusively study the test-time inference process in this work.
> Both (Chen et al., 2020) and (Chen et al., 2021) assume access to a reference classification model trained on real images (ImageNet).
> However, Chen et al. (2020) minimize the KL-divergence of the feature representations between the reference model and the model trained on synthetic data, while Chen et al. (2021) use a contrastive loss to facilitate model invariance to standard image augmentations (RandAugment).
> Chen et al. (2020) further seek layer-specific learning rates for improved generalization, whereas Chen et al. (2021) use the same learning rate for all layers, as usual.
>
> > Runtime.
>
> Although our test-time adaptation strategy is not real-time (yet), it nevertheless offers a flexible runtime-accuracy trade-off, which none of the previous methods provide.
> As we analyze in Fig. 2, IaBN already improves the baseline accuracy substantially at negligible computational costs.
> Complemented with Seg-TTT, the approach is still more efficient and accurate than model ensembles, which is a strong baseline established and widely used in the community.
>
> > IoU for MC-Dropout in Tab. 1(b).
>
> We only provided the ECE results in Tab. 1(b) to motivate Seg-TTT, which relies on improved model calibration.
> But in fact, IaBN also consistently improves the IoU of the MC-Dropout approach, which we report in the following table:
>
>  | Method |     CS     |     BDD    |      IDD    |
>  |-----------------|:--------:|:--------:|:--------:|
>  |**ResNet-50 (Baseline)**     | 30.95 | 28.52 | 32.78 |
>  |**w/ IaBN**     | 37.54 | 32.79 | 34.21 |
>  |**w/ MC-Dropout**     | 30.45 | 31.96 | 32.50 |
>  |**w/ IaBN + MC-Dropout**     | **38.84** | **35.13** | **35.55** |
>
> We will be happy to add these results to the revision, if advised by the reviewer.
>
> > Correctness.
>
> We kindly ask to specify which of our claims the reviewer deems incorrect or not well-supported; we will be happy to clarify.

---

### Author Response · Authors · 2021-11-16
**General response**

We sincerely thank reviewers for their time, the encouraging and thoughtful feedback.
We are delighted that the reviewers largely acknowledge the quality of writing and our extensive experimentation.
We have revised our paper following the reviewer suggestions.

We fully agree with reviewers **tfFH**, **YkXo**, and **FDdf** that many of the technical components that we study here already featured in previous works; in the text we point this out with appropriate references.
However, it is not the technical, but the *empirical novelty* that we believe our study delivers.
Through a thoroughly revised protocol for evaluating out-of-distribution robustness (cf. Sec. 5), our analysis challenges the yet mainstream assumption of a non-adaptive inference process at test time.
Nevertheless, even though our technical extension may seem incremental and somewhat trivial, its design is not coincidental, but is motivated by a careful analysis, as recognized by reviewer **Dswx**.
Importantly, such a test-time adaptation process has not been analyzed before, let alone as comprehensively, to our knowledge.
If a simple test-time adaptation approach arranged from familiar techniques consistently provides substantial accuracy benefits over the more sophisticated (and, at the time, technically novel) training strategies developed in the past 3 years, does it not still advance our understanding of model generalization?
We believe it does and we hope the reviewers agree.

---

### Decision · Program_Chairs · 2022-01-20

**Decision:**

Reject

**Comment:**

This submission presents a technique to improve generalization of urban scenes segmentation.  Based on a pre-trained deep net on synthetic data, the approach aims at adapting statistics on real target domain such as Cityscapes, BDD or IDD datasets using an Instance-adaptive Batch Normalization (IaBN) at test time. Results are reported on several synthetic to real scenarios.

Most of the reviewers were not convinced by the approach and have raised several issues. After rebuttal and discussion, no one really changed her/his mind. The novelty of the proposed method is limited to the use of the existing IaBN in this context except the one-sample adaptation.  Although the proposed method is effective on some benchmarks, the extra processing time may be a significant limitation. Additional comparisons are necessary. We encourage the authors to consider the reviewers feedbacks for future publication.